# HyenaMoE: A Hybrid and Scalable Architecture for Efficient Genomic Modeling

## Abstract

DNA sequences serve as the fundamental blueprint of cellular life, encoding critical information for gene regulation, protein synthesis, and a broad spectrum of essential biological processes. Owing to their sequential structure, DNA sequences bear similarities to natural language, motivating the adaptation of large language model architectures and the pretraining–finetuning paradigm in genomics. This has led to the emergence of genomic foundation models that perform well across a wide range of downstream tasks. Nonetheless, current approaches face structural limitations. Transformer-based models possess strong representational capacity for local contexts, making them well-suited for tasks involving short sequences. However, their scalability is limited by the quadratic complexity of attention mechanisms. In contrast, methods based on state space models offer high computational efficiency and can process long-range genomic inputs, but they generally perform less strongly than Transformer counterparts on shorter sequences. To address these limitations, we introduce HyenaMoE, a unified hybrid architecture designed for genomic modeling using 3-mer tokenization. HyenaMoE combines efficient HyenaLite blocks for long-range dependency modeling with attention layers enhanced by Mixture-of-Experts routing, enabling scalable capacity expansion and more efficient allocation of model resources across diverse inputs. This design supports a favorable balance between model expressiveness and computational efficiency. Experiments on three representative benchmarks demonstrate that HyenaMoE achieves state-of-the-art performance across a diverse array of genomic prediction tasks.

## 1 Introduction

Large-scale sequence models have driven rapid advancements in the field of machine learning, extending their impact beyond natural language processing (NLP) Achiam et al. (2023) into domains including mathematics, biology, and medicine. For instance, in the field of proteomics, large-scale sequence models have achieved remarkable success in protein structure prediction tasks. The AlphaFold series of models Senior et al. (2020); Jumper et al. (2021); Abramson et al. (2024), in particular, has demonstrated the ability to predict protein structures with high speed and accuracy, thereby accelerating drug discovery, deepening our understanding of disease mechanisms, and advancing developments in bioengineering.

However, within the central dogma, proteins represent only the final step. The central dogma describes the flow of genetic information from DNA to RNA through transcription, and from RNA to proteins through translation, ultimately giving rise to the fundamental functions of an organism Crick (1970). While proteins are the primary effectors of biological function, DNA serves as the foundational blueprint that encodes the full range of genetic instructions underlying cellular and organismal processes. As a result, large-scale sequence models have also become fundamental tools for DNA sequence analysis Zhou & Troyanskaya (2015). In contrast to protein sequences, DNA sequences contain extensive non-coding regions that play critical regulatory roles in gene expression, chromatin accessibility, and transcription factor binding. As a result, analyzing non-coding DNA has become a central focus of recent research, with growing interest in using large language models for tasks such as enhancer classification, promoter detection, and epigenetic mark prediction Ji et al. (2021); Zhou et al. (2023); Dalla-Torre et al. (2025).

A central challenge in the analysis of non-coding sequences lies in balancing input sequence length with model capacity. Longer contexts provide richer regulatory signals and generally improve task performance. As shown in Figure 1(**a**), extending the input length from 512 to 2,048 tokens yields a clear gain in classification accuracy across 17 genomic foundation models. However, scaling sequence length is nontrivial. Transformer-based models Vaswani et al. (2017), despite their strong representational power enabled by multi-head self-attention, suffer from quadratic complexity with respect to input length. This computational bottleneck restricts most existing models to pretraining on relatively short sequences (typically 512–4,096 tokens), covering only a small portion of the human genome. To address this limitation, efficient alternatives such as Hyena Poli et al. (2023) and Mamba Gu & Dao (2023) leverage long convolutions or state space models to reduce memory and computational cost. These designs can theoretically scale to million-token inputs under similar hardware constraints. Yet, as shown in Figure 1(**b**), although SSM-based models achieve competitive performance on certain tasks, they still fall short of Transformer counterparts on the majority of benchmarks, underscoring the trade-off between scalability and representational capacity.

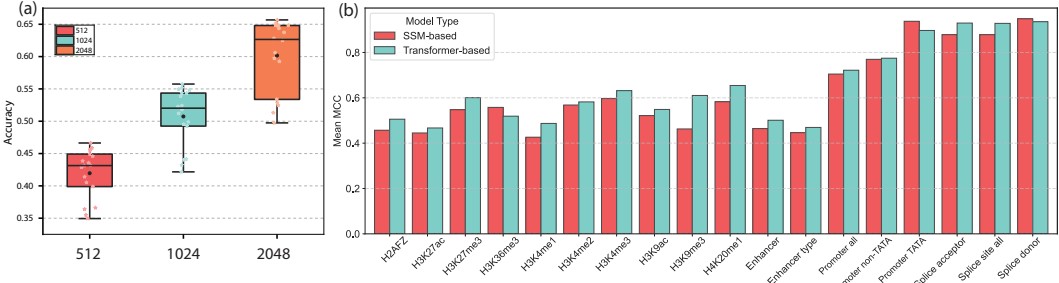

Figure 1: (**a**) Accuracy of 17 models on a species classification task across three input lengths (512, 1,024, 2,048 tokens). The figure illustrates qualitative scaling behavior — performance improves as models access more sequence context, serving as motivation rather than a biological claim. (**b**) Performance comparison of published models Wu et al. (2025) on the NT Benchmark. Transformer-based models outperform SSM-based models on 15 out of 18 tasks.

Hybrid architectures have emerged as a potential solution to address this trade-off. For example, in StripedHyena Nguyen et al. (2024), Transformer blocks are interleaved with Hyena blocks to construct a model that efficiently supports longer input sequences while maintaining strong sequence modeling capabilities. In StripedHyena2 Ku et al. (2025), the Hyena blocks are further refined into three types of convolutions: short explicit convolutions, medium regularized convolutions, and long implicit convolutions. By optimizing the arrangement of these components, StripedHyena2 achieves improved performance. However, models such as Evo and Evo2, which adopt hybrid architectures, are typically constructed with over one billion parameters. As a result, their strong performance may largely stem from scale rather than architectural design, and their effectiveness under comparable model sizes remains insufficiently validated.

Instead of attributing performance gains solely to larger model scale, a key question is how to expand model capacity in a more efficient manner. Mixture-of-Experts (MoE) architectures provide a compelling solution by employing conditional computation, where only a small subset of experts is activated for each input. This approach enables models to substantially increase their total parameter capacity without a proportional rise in computational cost Dai et al. (2024); Du et al. (2022); Zoph et al. (2022), making MoE a natural candidate for genomic sequence modeling.

To address the challenges outlined above, we propose HyenaMoE, a hybrid architecture that interleaves lightweight Hyena blocks (HyenaLite) with MoE-enhanced attention modules. HyenaLite ensures computational efficiency and effective long-range dependency modeling, while MoE expands model capacity through dynamic expert routing, enabling specialization across heterogeneous genomic patterns. This complementary design allows HyenaMoE to balance long-range efficiency with local adaptability, making it well-suited for the diverse demands of genomic modeling.

Our main contributions are as follows:

- We introduce HyenaLite, a lightweight variant of the Hyena operator that supports sliding-window inference with hierarchical time decomposition, achieving over 2× speedup and reduced memory usage compared to the original design.

- We adapt MoE to genomic sequence modeling, simplifying its design to align with our hybrid framework. This enables more efficient allocation of model capacity to distinct genomic elements such as promoters, enhancers, and non-coding regions.

- We present HyenaMoE, a parameter-efficient hybrid model that integrates HyenaLite for long-range dependencies and MoE-augmented attention for flexible local specialization, offering robust scalability from short to ultra-long input sequences.

- Extensive experiments across multiple benchmarks demonstrate that HyenaMoE consistently outperforms existing approaches, highlighting its advantages in both performance and efficiency.

## 2 PRELIMINARIES AND RELATED WORK

### 2.1 TRANSFORMERS AND ATTENTION

Transformer models (Vaswani et al., 2017) remain the dominant approach for genomic sequence modeling because self-attention captures rich local and mid-range dependencies. However, the $\mathcal{O}(L^2)$ cost with respect to sequence length $L$ fundamentally limits scalability to the 50k–1M bp regimes common in regulatory genomics. Recent genomic Transformers such as DNABERT (Ji et al., 2021), DNABERT-2 (Zhou et al., 2023), the Nucleotide Transformer series (Dalla-Torre et al., 2025), and large-scale models such as Gener*ator* (Wu et al., 2025) extend context windows through engineering improvements (e.g., FlashAttention, optimized activations). Yet, they still inherit attention's quadratic scaling and become expensive at longer contexts.

### 2.2 LONG-CONTEXT SEQUENCE MODELS: MAMBA AND HYENA

To address this limitation, recent architectures such as Mamba and Hyena replace pairwise attention with linear-time state-space or convolutional operators (Gu & Dao, 2023; Nguyen et al., 2023). These models process very long sequences efficiently but introduce their own trade-offs: while they scale to 100k–1M bp inputs, their purely convolutional or SSM structure often reduces local expressiveness and can underperform Transformers on motif- and boundary-sensitive regulatory tasks. Mamba-based models such as Caduceus (Schiff et al., 2024) and Hyena-based models such as HyenaDNA (Nguyen et al., 2023) demonstrate that increasing maximum context alone does *not* guarantee better predictive performance: despite supporting 131k–1M bp contexts, these models can lag behind Transformer-based baselines on key functional genomics tasks. These observations motivate hybrid architectures that combine the strengths of attention (local precision and expressiveness) with efficient long-range operators (scalability), rather than relying on either mechanism alone.

### 2.3 HYBRID ARCHITECTURES

StripedHyena and StripedHyena2 combine the Hyena operator, a hierarchy of implicit long convolutions, with grouped query attention and gated MLPs to efficiently model both local and global dependencies under subquadratic complexity. Layers alternate between attention and Hyena blocks to balance expressiveness and scalability. StripedHyena2 further improves the architecture through low rank projections and adaptive routing Zhou et al. (2022). Building on these designs, StripedHyena7B is a 7B parameter model that eliminates full attention layers entirely, relying on grouped query attention Ainslie et al. (2023) and Hyena blocks. Trained on sequences up to 32k tokens, it achieves more than a 2x speedup in training and inference at 128k context lengths compared to strong Transformer baselines.

The Evo series Nguyen et al. (2024); Brixi et al. (2025) extends the StripedHyena family to genomics, applying hybrid architectures to single nucleotide resolution inputs up to 650k tokens. These models demonstrate that compositional designs unifying state space models, convolution, and attention can generalize beyond natural language tasks, offering scalable and expressive solutions for highly structured biological data.

### 2.4 MIXTURE OF EXPERTS

MoE models Shazeer et al. (2017) increase model capacity by activating only a sparse subset of specialized expert networks for each input. Formally, given $x \in \mathbb{R}^D$, an MoE layer outputs:

$$y = \sum_{i=1}^{E} g_i(x) f_i(x), \qquad (1)$$

where $E$ denotes the number of experts, $f_i(x)$ is the $i$-th expert, and $g_i(x)$ is the gating function, typically a softmax. MoE achieves computational efficiency by routing each input through a few experts, but challenges such as load balancing and routing stability remain active research topics Lepikhin et al. (2020); Fedus et al. (2021). GShard Lepikhin et al. (2020) and Switch Transformer Fedus et al. (2021) demonstrate that MoE-based architectures can scale to hundreds of billions of parameters while maintaining computational efficiency through sparse activation. More recent variants extend MoE to vision and multi-domain task learning in NLP, as exemplified by V-MoE Riquelme et al. (2021), MoDE Schafhalter et al. (2024), and DeepSeekMoE Dai et al. (2024), thereby demonstrating its versatility across modalities and task distributions. These developments establish MoE as a foundational block in the design of scalable and computationally efficient deep neural networks.

## 3 METHODOLOGY

Our architecture is built around the multi-scale structure of genomic sequences. Attention layers handle short-range motif and boundary features, HyenaLite captures long-range dependencies with subquadratic cost, and MoE-augmented FFN blocks provide conditional capacity for heterogeneous genomic contexts. This division of labor forms a principled hybrid model rather than a collection of assembled modules. In this section, we begin by introducing the optimized HyenaLite block, which enhances the original design through targeted implementation-level improvements for efficient long-range dependency modeling (Sec. 3.1). Next, we describe our improved Transformer block with integrated MoE-augmented FFN, which increases model capacity via conditional computation while preserving computational efficiency (Sec. 3.2). Finally, we present the overall framework and training pipeline, outlining the architectural composition and the full workflow for pretraining and downstream finetuning (Sec. 3.3.1).

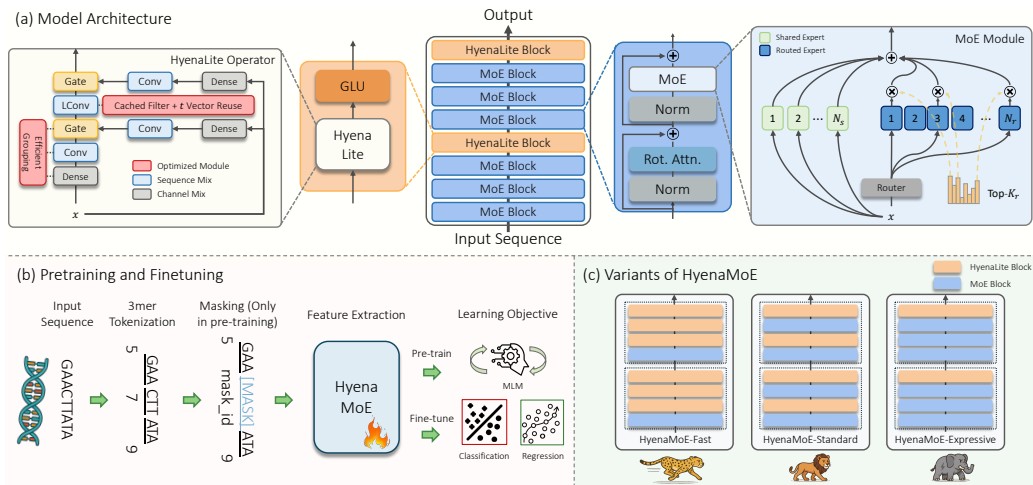

Figure 2: Overview of the HyenaMoE framework. **(a)** Model architecture combining HyenaLite blocks and MoE-augmented attention for scalable, efficient sequence modeling. HyenaLite uses grouped depthwise convolutions and cached filters for fast temporal mixing; the MoE-augmented FFN mixes shared and routed experts with top-$K_r$ gating. **(b)** Pretraining and finetuning pipeline: genomic sequences are 3-mer tokenized, masked (pretraining only), and encoded by HyenaMoE. The model is trained with MLM and adapted to classification/regression tasks. **(c)** Three variants offer capacity-computation trade-offs: Fast (light), Standard (balanced), and Expressive (MoE-rich). Keys: GLU – Gated Linear Unit; Norm – Normalization; Rot. Attn. – Rotary Attention.

### 3.1 HYENALITE BLOCK

To improve the efficiency of long-range modeling, we introduce the HyenaLite Block, a streamlined variant of the original Hyena operator that delivers higher throughput and a lower memory footprint.

HyenaLite maintains the expressive filtering capacity of its predecessor while integrating a number of architectural refinements that enhance scalability and deployment performance.

Given an input sequence $x \in \mathbb{R}^{B \times L \times D}$, the block begins by normalizing and projecting the input into three parallel streams:

$$z = \text{Linear}(\text{RMSNorm}(x)), \quad z \in \mathbb{R}^{B \times L \times 3D}. \tag{2}$$

This is split into $[x_2, x_1, v] \in \mathbb{R}^{B \times L \times D}$, which are passed into a parameterized convolution-based filter to capture long-range dependencies. The filter operates on a combination of multiplicative and additive interactions:

$$y(t) = x_2(t) \odot \left[ (h * (x_1 \odot v))(t) + \omega_D \odot x_1(t) \odot v(t) \right], \tag{3}$$

where all products $\odot$ denote element-wise multiplication, $\omega_D \in \mathbb{R}^D$ is a learnable residual weight, the filter kernel $h \in \mathbb{R}^{1 \times D \times L}$ is constructed via a sum of complex exponentials:

$$h_d(t) = \text{Re}\left( \sum_{k=1}^{S} r_{d,k} e^{\lambda_{d,k} t} \right). \tag{4}$$

To reduce runtime overhead, HyenaLite caches the computed filter kernel $h$ during inference, avoiding redundant recomputation. Moreover, the time index vector $t$ used to parameterize the filter is shared and reused across all layers, eliminating per-layer positional computation. The block also employs efficient grouped convolutions to improve memory locality and enable parallelism across channels.

The full block is wrapped in a residual structure and finalized with a lightweight gated MLP:

$$\text{HyenaLiteBlock}(x) = \text{MLP}(\text{RMSNorm}(x + \text{HyenaFilter}(z))).$$

These optimizations significantly reduce computational and memory costs, enabling high-throughput modeling of long sequences without compromising the temporal modeling capacity inherent to the Hyena framework.

To support inference on extremely long sequences, HyenaLite employs a sliding-window mechanism. During inference, the input is partitioned into overlapping windows, and each HyenaLite layer maintains a compact cache of FIR/IIR convolutional states. When the window advances, these states are updated by a single recurrent step and reused for the next window, eliminating the need to recompute activations over the entire history. Consequently, both compute and memory scale with network depth rather than total sequence length, enabling efficient inference on 100k–150k bp inputs with minimal degradation.

## 3.2 MoE-augmented Transformer Block

Inspired by DeepSeekMoE (Dai et al., 2024), we adopt a dual-pathway MoE-augmented FFN mechanism in our Attention block to balance expressivity, routing flexibility, and computational efficiency. Similar to DeepSeekMoE, our architecture incorporates both *shared experts* that process all tokens and *routed experts* that are dynamically selected for each token via a learnable gating function.

Our design departs from DeepSeekMoE in several aspects. Notably, instead of relying on explicit auxiliary load-balancing losses, we adopt a simplified routing strategy based on score normalization and token-wise top-$k$ expert selection. This design implicitly encourages balanced expert utilization during training without introducing additional optimization objectives. We also implement expert dispatch and computation in a communication-efficient distributed manner.

Let $\mathbf{u}_t \in \mathbb{R}^D$ denote the FFN input of the $t$-th token. The output $\mathbf{h}'_t$ is computed as:

$$\mathbf{h}'_t = \mathbf{u}_t + \sum_{i=1}^{N_s} \text{FFN}_i^{(s)}(\mathbf{u}_t) + \sum_{i=1}^{N_r} g_{i,t} \cdot \text{FFN}_i^{(r)}(\mathbf{u}_t), \tag{5}$$

where $N_s$ and $N_r$ are the number of shared and routed experts, respectively. Each shared expert $\text{FFN}_i^{(s)}$ operates unconditionally, while routed experts are activated based on token-specific affinity scores produced by a gating network. The affinity between token $t$ and expert $i$ is defined as:

$$s_{i,t} = \sigma(\mathbf{u}_t^\top \mathbf{e}_i), \tag{6}$$

where $\sigma$ is the sigmoid function, and $\mathbf{e}_i$ is a learnable key vector associated with expert $i$. Based on these scores, the top-$K_r$ experts are selected, and the normalized gating value is given by:

$$g_{i,t} = \frac{s'_{i,t}}{\sum_{j=1}^{N_r} s'_{j,t}} \quad \text{where} \quad s'_{i,t} = \begin{cases} s_{i,t}, & s_{i,t} \in \text{Topk}(\{s_{j,t}\}_{j=1}^{N_r}, K_r) \\ 0, & \text{otherwise} \end{cases}. \quad (7)$$

This top-$k$ routing is implemented efficiently in parallel across tokens and experts. Rather than incorporating additional loss terms for load balancing, our routing framework adapts dynamically through learnable score scaling and normalized expert selection.

Furthermore, in contrast to DeepSeekMoE which adds explicit bias terms $b_i$ to affinity scores and adjusts them during training, our method achieves implicit load balancing through sparse activation and modular expert updates. Empirically, this design provides comparable performance and utilization uniformity while significantly reducing implementation complexity.

All experts are implemented as gated MLPs with the following form:

$$\text{Expert}(x) = \mathbf{W}_2(\text{SiLU}(\mathbf{W}_1 x) \odot \mathbf{W}_3 x), \quad (8)$$

where $\odot$ denotes element-wise multiplication and all $\mathbf{W}_i$ are learnable projection matrices. Shared experts are vectorized as a single MLP layer, while routed experts are distributed across devices to support data-parallel execution. This structure enables conditional compute at scale, maintaining both high throughput and balanced expert engagement.

### 3.3 HYENAMoE

#### 3.3.1 OVERALL FRAMEWORK OF HYENAMoE

We propose HyenaMoE, a hybrid architecture that interleaves convolutional HyenaLite blocks with MoE-augmented attention modules. This design couples the computational efficiency of HyenaLite with the conditional expressiveness of MoE-augmented FFN, giving the model flexible trade-offs between scalability and representational capacity.

By adjusting the ratio and arrangement of HyenaLite and MoE blocks, HyenaMoE can be instantiated into three variants, each optimized along a different point of the efficiency–capacity spectrum.

- **HyenaMoE-Fast** uses one MoE-augmented attention and three HyenaLite layers per four-block group, emphasizing inference speed and memory efficiency. This makes it suitable for ultra-long sequence tasks such as genome-scale scanning, where throughput outweighs representational richness.

- **HyenaMoE-Standard** alternates HyenaLite and MoE-augmented attention layers within each group, striking a balance between modeling capacity and efficiency. As the default configuration in our benchmarks, it supports both long-context processing and expressive learning across diverse tasks.

- **HyenaMoE-Expressive** assigns three MoE-augmented attention layers and one HyenaLite layer per group, prioritizing representational capacity for complex tasks like fine-grained genomic annotation. While more compute-intensive, it benefits from the flexibility and depth of expert attention.

Together, these variants define a unified architectural design space that enables principled customization. HyenaMoE can be scaled along axes of performance and efficiency without necessitating structural redesign, making it broadly applicable to a wide range of biological modeling tasks.

#### 3.3.2 TRAINING PARADIGM

Our training pipeline adopts a standard pretrain-then-finetune paradigm. During pretraining, we use the same large-scale genomic sequence corpus as DNABERT-2 (Zhou et al., 2023), which consists of 2.75B bases from the human genome and a multi-species genome dataset comprising 32.49B bases from 135 species across six categories. We construct 2,000 bp pre-training segments by concatenating adjacent genomic fragments; under 3-mer tokenization this yields 667 tokens, which are then packed into fixed 1,024-token windows for batching. All downstream tasks share the same pretrained checkpoint; only task-specific heads and fine-tuning differ. Following DNABERT-2, we

exclude sequences containing ambiguous bases (N) and retain only canonical nucleotides (A, T, C, G). To encode input sequences, we adopt a 3-mer tokenization strategy, which expands the effective token vocabulary while preserving biological relevance. The model is trained using a masked language modeling (MLM) objective, enabling it to learn contextual representations of DNA sequences across diverse genomic regions and regulatory landscapes. For downstream evaluation, we perform full-parameter finetuning on task-specific datasets, allowing HyenaMoE to adapt to the specific requirements of individual biological prediction tasks while preserving the generalization capabilities acquired during pretraining. The architecture supports a broad range of tasks, including sequence-level classification, span-based prediction, and sequence regression, with a unified backbone used across all stages to ensure consistent and transferable representations.

## 4 EXPERIMENTS

### 4.1 BASELINE

We compare our method against recent genomic foundation models spanning three architectural categories: SSM-based (HyenaDNA, Caduceus), Transformer-based (DNABERT, DNABERT-2, NT-v2, Gener*ator*), and Hybrid (Evo2). These baselines vary in their tokenization strategies, model architectures, training objectives, and supported input lengths. To ensure a fair comparison, all models are finetuned in a full-parameter manner, except for Evo2, which is evaluated using a linear probing protocol Zhai et al. (2022) due to its large scale and computational overhead. Full implementation details and training configurations are provided in the appendix.

### 4.2 SETUP AND METRIC

We evaluate our proposed model from two complementary perspectives: downstream performance and computational efficiency. For downstream evaluation, we conduct experiments on three representative benchmarks: Genome Understanding Evaluation (GUE) Benchmark, Nucleotide Transformer Tasks, and Genomics Benchmark. Since all tasks are classification-based, we report Accuracy as the primary metric in the main paper, with detailed Macro-F1 and Matthews Correlation Coefficient Matthews (1975) results provided in the appendix.

To assess computational efficiency, we benchmark average latency, throughput, peak memory usage, and FLOPs across different architectural variants, including dense Transformer and hybrid configurations. FLOPs are computed on input sequences of length 2,048, following common practice in genomic analysis. Unless otherwise specified, we employ non-parametric bootstrapping Tibshirani & Efron (1993) with 1,000 resamples to estimate the standard deviation (std) for all reported results. For each evaluation, the best-performing checkpoint on the validation set is selected for testing.

Across all experiments, we keep the majority of hyperparameters (e.g., learning rate, batch size, weight decay) fixed across datasets. A detailed description of dataset configurations, training hyperparameters, evaluation scripts, and environment settings is provided in the appendix to ensure full reproducibility.

### 4.3 PERFORMANCE EVALUATION

Table 1 presents a comprehensive comparison of ten representative genomic foundation models evaluated on three benchmark suites: GUE, Nucleotide Transformer Tasks, and Genomics Benchmark. Performance is reported in terms of average classification accuracy with standard deviation (computed via 1,000 bootstrap resamples). Our proposed model, HyenaMoE-E, consistently achieves strong results across all benchmarks. In the GUE Benchmark, HyenaMoE-E achieves the highest accuracy on 5 out of 7 tasks, including the most challenging datasets such as TF-H and SSP, which require modeling subtle transcription factor binding and regulatory signals. It also performs competitively on the remaining tasks, ranking in the top two on all. Within Nucleotide Transformer Tasks, which consists of 17 tasks covering histone modifications, enhancer/promoter activity, and splice sites, HyenaMoE-E ranks first on 9 tasks and second on 6, demonstrating robust performance across both short-range and long-range regulatory elements. Notably, it achieves top scores on biologically complex targets such as H3K27ac, H3K9ac and Enhancer Types, underscoring its generalization ability across diverse sequence contexts. In Genomics Benchmark, HyenaMoE-E leads on 6 out of 8 tasks and ranks second on the remaining two. It achieves particularly strong performance on difficult classification settings such as Human vs Worm, Human Enhancers Ensembl, and Human Regulatory, where it either achieves or matches the best reported accuracy. The green highlights in

Table 1 indicate that HyenaMoE outperforms all other models of comparable size, demonstrating strong efficiency and performance per parameter.

Table 1: Performance comparison across three genomic benchmarks: GUE Benchmark, Nucleotide Transformer Tasks, and Genomics Benchmark. Each cell reports the average accuracy, with standard deviation ($\pm$) estimated from 1,000 bootstrap resamples. Models are categorized into SSM-based, Transformer-based, and Hybrid groups. The best is **bolded**, and the second best is underlined. Green highlighting indicates models that belong to the same parameter scale group, allowing for fair comparisons across similar-sized architectures.

| Model / Task | SSM-based | | Transformer-based | | | | Hybrid | | | |
|---|---|---|---|---|---|---|---|---|---|---|
| | HyenaDNA (436K) | Caduceus (470K) | DNABERT (86M) | DNABERT-2 (117M) | NT-v2 (500M) | Generator (1.2B) | Evo2 (7B) | HyenaMoE-F (116M) | HyenaMoE-S (131M) | HyenaMoE-E (145M) |
| *GUE Benchmark* | | | | | | | | | | |
| CPD | 0.807±0.009 | 0.805±0.009 | 0.841±0.008 | 0.791±0.009 | 0.844±0.013 | 0.819±0.009 | 0.697±0.010 | 0.832±0.008 | 0.840±0.007 | **0.845±0.009** |
| CVC | 0.278±0.005 | 0.231±0.004 | 0.496±0.005 | 0.673±0.005 | 0.642±0.005 | 0.666±0.005 | 0.123±0.003 | 0.670±0.003 | **0.673±0.004** | 0.672±0.003 |
| EMP | 0.747±0.008 | 0.754±0.008 | 0.733±0.008 | 0.751±0.008 | 0.778±0.007 | 0.784±0.008 | 0.725±0.008 | 0.790±0.009 | **0.798±0.008** | 0.797±0.009 |
| PD | 0.879±0.008 | 0.854±0.008 | 0.902±0.007 | 0.868±0.010 | 0.904±0.007 | **0.923±0.007** | 0.779±0.010 | 0.901±0.009 | 0.910±0.009 | 0.905±0.008 |
| SSP | 0.567±0.007 | 0.566±0.007 | 0.879±0.005 | 0.843±0.001 | **0.942±0.006** | 0.937±0.004 | 0.568±0.007 | 0.920±0.005 | 0.938±0.005 | 0.929±0.006 |
| TF-H | 0.803±0.012 | 0.817±0.012 | 0.778±0.013 | 0.855±0.009 | **0.901±0.005** | 0.822±0.012 | 0.674±0.015 | 0.860±0.015 | 0.866±0.012 | 0.867±0.013 |
| TF-M | 0.786±0.016 | 0.806±0.015 | 0.729±0.017 | 0.766±0.009 | 0.758±0.009 | 0.837±0.014 | 0.638±0.018 | 0.839±0.019 | 0.844±0.015 | **0.849±0.018** |
| *Nucleotide Transformer Tasks* | | | | | | | | | | |
| H2AFZ | **0.752±0.008** | 0.710±0.008 | 0.728±0.008 | 0.704±0.008 | 0.731±0.008 | 0.722±0.008 | 0.680±0.008 | 0.734±0.011 | 0.737±0.009 | 0.744±0.008 |
| H3K27ac | 0.741±0.011 | 0.720±0.011 | 0.717±0.012 | 0.721±0.009 | 0.737±0.011 | 0.742±0.011 | 0.646±0.012 | 0.731±0.011 | 0.732±0.012 | **0.759±0.012** |
| H3K27me3 | 0.780±0.007 | 0.774±0.008 | 0.782±0.008 | 0.757±0.016 | 0.788±0.008 | **0.804±0.007** | 0.741±0.008 | 0.789±0.011 | 0.790±0.008 | 0.798±0.008 |
| H3K36me3 | 0.772±0.008 | 0.764±0.008 | 0.768±0.008 | 0.773±0.008 | 0.804±0.008 | **0.805±0.008** | 0.684±0.008 | 0.797±0.008 | 0.791±0.008 | 0.796±0.008 |
| H3K4me1 | 0.730±0.008 | 0.711±0.008 | 0.716±0.008 | 0.712±0.012 | **0.736±0.008** | 0.736±0.008 | 0.664±0.009 | 0.725±0.009 | 0.725±0.009 | 0.726±0.008 |
| H3K4me2 | 0.759±0.009 | 0.733±0.009 | 0.753±0.009 | 0.743±0.009 | 0.763±0.009 | 0.768±0.009 | 0.707±0.009 | **0.782±0.010** | 0.782±0.010 | 0.781±0.009 |
| H3K4me3 | 0.815±0.014 | 0.806±0.014 | 0.812±0.014 | 0.816±0.008 | 0.818±0.014 | 0.807±0.014 | 0.748±0.016 | 0.852±0.017 | **0.853±0.015** | 0.853±0.013 |
| H3K9ac | 0.768±0.013 | 0.767±0.014 | 0.731±0.015 | 0.765±0.008 | 0.772±0.013 | 0.751±0.014 | 0.708±0.014 | 0.765±0.009 | 0.757±0.015 | **0.774±0.014** |
| H3K9me3 | 0.700±0.016 | 0.691±0.016 | 0.700±0.016 | 0.703±0.014 | 0.705±0.015 | 0.720±0.016 | 0.569±0.017 | 0.704±0.009 | 0.647±0.017 | **0.726±0.015** |
| H4K20me1 | 0.802±0.008 | 0.790±0.008 | 0.801±0.008 | 0.803±0.008 | **0.822±0.008** | 0.813±0.008 | 0.753±0.008 | 0.806±0.009 | 0.808±0.009 | 0.806±0.008 |
| Enhancers | 0.740±0.008 | 0.738±0.008 | 0.743±0.008 | 0.742±0.008 | 0.760±0.008 | 0.736±0.008 | 0.702±0.008 | 0.727±0.008 | 0.736±0.009 | **0.762±0.008** |
| Enhancer types | 0.710±0.008 | 0.707±0.008 | 0.706±0.008 | 0.700±0.016 | 0.714±0.008 | 0.720±0.008 | 0.679±0.009 | 0.700±0.008 | 0.704±0.009 | **0.725±0.009** |
| Promoter All | 0.858±0.009 | 0.857±0.009 | 0.869±0.009 | 0.866±0.009 | 0.879±0.009 | 0.883±0.009 | 0.846±0.009 | 0.906±0.008 | **0.911±0.009** | 0.884±0.009 |
| Promoter Non-TATA | 0.872±0.009 | 0.872±0.009 | 0.864±0.009 | 0.873±0.009 | 0.879±0.009 | 0.886±0.009 | 0.862±0.009 | 0.890±0.010 | **0.899±0.010** | 0.899±0.009 |
| Promoter TATA | 0.896±0.020 | 0.860±0.023 | **0.902±0.020** | 0.858±0.016 | 0.897±0.021 | 0.868±0.024 | 0.812±0.027 | 0.851±0.025 | 0.854±0.029 | 0.861±0.027 |
| Splice sites Acceptor | 0.848±0.007 | 0.792±0.007 | **0.969±0.003** | 0.828±0.009 | 0.936±0.004 | 0.884±0.006 | 0.708±0.008 | 0.828±0.004 | 0.812±0.009 | 0.807±0.008 |
| Splice sites All | 0.749±0.008 | 0.571±0.009 | **0.966±0.003** | 0.734±0.010 | 0.942±0.004 | 0.898±0.005 | 0.483±0.009 | 0.742±0.005 | 0.767±0.009 | 0.744±0.009 |
| Splice sites Donor | 0.832±0.007 | 0.797±0.007 | **0.980±0.003** | 0.835±0.019 | 0.952±0.004 | 0.914±0.005 | 0.722±0.008 | 0.763±0.008 | 0.682±0.009 | 0.772±0.008 |
| *Genomics Benchmark* | | | | | | | | | | |
| Coding vs Intergenomic | 0.906±0.002 | 0.905±0.002 | 0.937±0.002 | 0.895±0.002 | 0.946±0.001 | **0.957±0.001** | 0.908±0.002 | 0.915±0.004 | 0.920±0.003 | 0.956±0.002 |
| Human vs Worm | 0.961±0.001 | 0.961±0.001 | 0.958±0.001 | 0.947±0.001 | 0.970±0.001 | 0.975±0.001 | 0.714±0.001 | 0.969±0.003 | 0.979±0.002 | **0.985±0.001** |
| Mouse Enhancers | 0.772±0.027 | 0.500±0.031 | 0.782±0.026 | 0.810±0.025 | 0.815±0.024 | 0.823±0.025 | 0.711±0.029 | 0.825±0.023 | **0.831±0.028** | 0.831±0.029 |
| Human Enhancers Cohn | 0.728±0.005 | 0.736±0.005 | 0.700±0.005 | 0.500±0.005 | 0.747±0.005 | 0.751±0.006 | 0.706±0.005 | 0.742±0.006 | 0.751±0.006 | **0.752±0.005** |
| Human Enhancers Ensembl | 0.838±0.002 | 0.779±0.002 | 0.845±0.002 | 0.766±0.002 | 0.857±0.002 | **0.879±0.002** | 0.676±0.002 | 0.840±0.003 | 0.846±0.003 | 0.849±0.002 |
| Human Regulatory | 0.870±0.001 | 0.844±0.002 | 0.936±0.001 | 0.679±0.002 | 0.940±0.001 | 0.921±0.001 | 0.538±0.002 | 0.939±0.002 | 0.942±0.002 | **0.959±0.002** |
| Human nonTATA Promoters | 0.925±0.003 | 0.842±0.004 | 0.878±0.003 | **0.946±0.002** | 0.896±0.003 | 0.921±0.003 | 0.788±0.004 | 0.928±0.004 | 0.938±0.004 | 0.935±0.004 |
| Human OCR Ensembl | 0.765±0.002 | 0.719±0.003 | 0.747±0.002 | 0.511±0.003 | 0.745±0.002 | 0.779±0.002 | 0.575±0.003 | 0.778±0.003 | **0.789±0.005** | 0.788±0.003 |

We evaluated the average performance of all models across tasks and found that HyenaMoE-E achieved the highest overall accuracy (82.87%), followed closely by Gene*rator* (82.79%) and HyenaMoE-S (82.78%). NT-v2 and HyenaMoE-F also performed competitively, with average scores of 82.57% and 82.55%, respectively, which reflects the strong and consistent effectiveness of the HyenaMoE family across diverse genomic tasks. Although HyenaMoE-F ranked slightly lower in overall average accuracy, it demonstrated clear advantages in long-context modeling. In a regression task predicting bulk RNA expression across varying input lengths, HyenaMoE-F consistently outperformed all baselines, including NT-v2, by achieving higher $R^2$ scores across all tested lengths from 512 to 8,192. This suggests superior scalability and a strong ability to capture long-range dependencies, which are critical in many biological applications. These results confirm the robustness and efficiency of the hybrid HyenaMoE design. Further details are provided in the appendix.

Table 2: LRB results under 32k–128k bp contexts. We report representative metrics from the gene-regulation (enhancer–gene, promoter–gene) and chromatin prediction (histone marks) categories, which are specifically designed to evaluate long-range dependencies. Full results for all downstream variant-effect, gene-regulation, and chromatin tasks are provided in the Supplementary Materials.

| Model | Enhancer (AUROC) | Promoter (AUPRC) | Histone (AUPRC) |
|---|---|---|---|
| NTv2-500M (12k) | 0.82 ± 0.002 | 0.79 ± 0.006 | 0.38 ± 0.003 |
| HyenaMoE (12k) | 0.81 ± 0.003 | 0.80 ± 0.007 | 0.38 ± 0.004 |
| HyenaMoE (15k) | **0.82 ± 0.003** | **0.81 ± 0.006** | **0.39 ± 0.004** |

As shown in Table 2, we evaluate HyenaMoE on the Human Genomics Long-Range Benchmark (LRB), which probes enhancer–gene interactions, promoter–gene mapping, and long-range chromatin prediction under 32k–128k bp contexts. These tasks require models to capture regulatory dependencies spanning tens to hundreds of kilobases. HyenaMoE achieves performance on par with, and in several cases exceeding, the 12k-context NTv2-500M baseline, and extending the effective

context from 12k to 15k yields consistent improvements across all task categories. The results align closely with our >100 kbp sliding-window inference experiments, where HyenaLite maintains stable accuracy on 100k–150k bp regions with minimal degradation. Together, these findings confirm that HyenaMoE effectively models long-range genomic structure and benefits from longer contexts without sacrificing stability.

Table 3: Ablation study of different architectural components. We report average classification accuracy (mean ± std) across three representative benchmarks.

| Model | GUE Benchmark | Nucleotide Transformer Tasks | Genomics Benchmark |
|---|---|---|---|
| HyenaLite-only | $0.810 \pm 0.008$ | $0.751 \pm 0.009$ | $0.823 \pm 0.007$ |
| Attention-only | $0.824 \pm 0.010$ | $0.783 \pm 0.008$ | $0.855 \pm 0.006$ |
| Hybrid (no MoE) | $0.831 \pm 0.007$ | $0.784 \pm 0.009$ | $0.875 \pm 0.008$ |
| HyenaMoE-S | $\mathbf{0.838 \pm 0.006}$ | $\mathbf{0.790 \pm 0.007}$ | $\mathbf{0.882 \pm 0.006}$ |

Table 3 shows that HyenaLite ensures efficiency and strong long-range dependency modeling, but its expressiveness is limited for diverse local genomic patterns. MoE-augmented FFN enhances scalability through dynamic expert routing, enabling specialization across heterogeneous genomic signals such as promoters and enhancers. By combining the two, HyenaMoE balances long-range efficiency with local adaptability, offering broader coverage of genomic tasks from short-sequence classification to ultra-long context modeling.

### 4.4 TOKENIZATION CHOICE

Genomic sequences exhibit structure at multiple resolutions, making the choice of tokenizer an important design decision. We compare five tokenization schemes under controlled settings: fixed $k$-mers ($k \in \{3, 4, 5, 6\}$) and a DNA-specific BPE vocabulary. All pretraining runs use the same model size, number of updates, and *base-level* context length to ensure fair comparison.

Across the three benchmark suites (GUE, NT tasks, and Genomics Bench), 3-mer consistently provides the best overall performance, particularly on motif- and boundary-sensitive tasks. Larger $k$ (5- or 6-mer) and BPE tokenization shorten the token sequence and therefore cover more bases within a fixed token budget, offering slight advantages on NT-style long-range tasks but at the cost of reduced motif resolution. Full quantitative results, including all metrics and variance estimates, are reported in the Supplementary Materials.

### 4.5 EFFICIENCY EVALUATION

We evaluate the computational efficiency of six representative architectures: Dense Attention, Pure Hyena, StripedHyena, and three variants of HyenaMoE (Fast, Standard, and Expressive). We compare their per-token FLOPs across varying input sequence lengths. Dense Attention incurs rapidly increasing cost due to its quadratic complexity, making it inefficient for long-context tasks. Pure Hyena, composed entirely of Hyena layers, maintains constant FLOPs per token and serves as a theoretical lower bound. StripedHyena adds a small number of attention layers to a Hyena backbone, balancing scalability and modeling capacity. HyenaMoE variants further enhance performance through MoE-augmented FFN while preserving the efficient FLOPs scaling of Hyena-based architectures. Even the most expressive variant remains far more efficient than dense attention on long sequences.

We also compare the inference efficiency of StripedHyena and HyenaMoE across a range of batch sizes, focusing on latency, throughput, and peak memory usage. Both models share the same core architecture, but HyenaMoE incorporates an MoE-augmented FFN mechanism for enhanced expressiveness. At small batch sizes, HyenaMoE introduces moderate overhead in latency and memory due to expert routing. However, as batch size increases, this overhead becomes negligible relative to the total computation. Notably, HyenaMoE sustains higher throughput than StripedHyena at large batch sizes while maintaining similar memory efficiency. This indicates that HyenaMoE scales more favorably under parallel workloads, offering improved utilization without sacrificing performance, which makes it especially suitable for high-throughput inference scenarios.

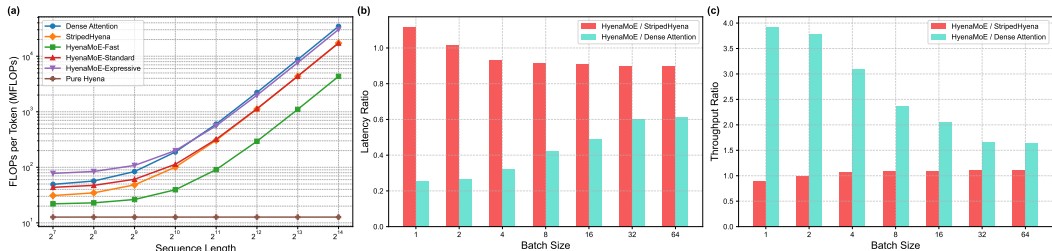

Figure 3: Comparison of computational efficiency and scalability. **(a)** FLOPs per token vs. sequence length shows Hyena's constant-time complexity, with HyenaMoE offering a balanced trade-off. **(b)** Latency and **(c)** throughput measured at 2,048-token sequences, where values shown as "A/B" denote ratios relative to a dense-attention baseline. This setting serves as an operator-level long-sequence stress test rather than a downstream task evaluation, highlighting HyenaMoE's efficiency (up to 3.9× higher throughput).

### 4.6 MIXTURE-OF-EXPERTS ABLATIONS

We conduct a series of ablations to understand how sparse routing, expert count, and gating normalization contribute to the performance and stability of HyenaMoE. Across all analyses, we find that model improvements stem from stable conditional computation rather than parameter scaling.

**Expert-load stability.** We measure routing balance using the Max/Avg load ratio, the coefficient of variation (CV), and the number of inactive ("dead") experts. Without score normalization, routing becomes highly unstable (Max/Avg $\approx 7.2$, CV $\sim 10^5$) and 12–20 experts collapse during training. Adding a Switch-like balancing loss does not improve stability. In contrast, score normalization—with or without the balancing loss—maintains well-balanced utilization (Max/Avg $\approx 3.1$–$3.5$), low CV, and zero dead experts in both pre-training and fine-tuning. These results establish score-normalized gating as necessary and sufficient for stable MoE behavior.

**Expert count.** Increasing the number of experts from 16 to 32 to 64 yields consistent gains in accuracy ($83.4\% \rightarrow 84.7\% \rightarrow 85.9\%$), higher routing entropy, and reduced load imbalance. Beyond 64 experts, performance saturates, suggesting that improvements arise from functional diversity rather than increased parameter counts.

**Routing sparsity and gating strategies.** We compare Top-1 and Top-2 routing under three gating variants: (1) no normalization, (2) Switch-like loss only, and (3) score normalization. Without normalization, both routing schemes degrade (81–82% accuracy) and produce many dead experts. Score-normalized gating is the only setting that achieves the highest accuracy (85.9%) while maintaining zero dead experts under Top-2 routing, confirming that stable sparse routing—not stochasticity or smoothing—drives performance gains.

## 5 CONCLUSION

In this work, we introduce HyenaMoE, a hybrid and scalable architecture for efficient genomic modeling. HyenaMoE integrates lightweight HyenaLite operators with sparse MoE-augmented attention, jointly leveraging the inductive bias of convolutional filters and the dynamic capacity of expert routing. Addressing the limitations of existing genomic models in scalability and efficiency, our design achieves state-of-the-art performance across diverse benchmarks while substantially reducing parameter count and pretraining cost. We further examine the architectural flexibility of HyenaMoE by varying the relative contributions of its attention and convolutional pathways, enabling adaptable variants tailored to different input scales and task complexities. These findings highlight the potential of hybrid modeling as a robust paradigm for efficient genomic sequence analysis, with future extensions to ultra-long sequence contexts representing an important direction for continued research.

## 6 ETHICS STATEMENT

This work does not involve human subjects, sensitive personal data, or any potentially harmful applications. All datasets used are publicly available and widely adopted in the research community. We adhered to responsible research practices, including proper citation of prior work and transparent reporting of experimental details. The methods proposed in this paper are intended solely for scientific and educational purposes, and we do not foresee any direct misuse or ethical risks associated with their application.

## 7 REPRODUCIBILITY STATEMENT

We have taken several steps to ensure the reproducibility of our results. Detailed descriptions of model architectures, training procedures, and hyperparameters are included in the main text and appendix. Additional implementation details are provided in the supplementary material. All datasets used are publicly accessible, and we describe preprocessing steps thoroughly. The source code will be released via a public repository after the paper is accepted, in order to facilitate replication of our experiments.

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
