# A Appendix

## A.1 Model Configurations for Fig. 1(a)

The following models were used to generate the results shown in **Fig. 1(a)** of the main paper. We provide the list of model names used, with learning rate suffixes removed for clarity.

- DNABERT-3
- DNABERT-4
- DNABERT-5
- DNABERT-6
- GENA-LM-BERT-Base-LastLN-T2T
- GENA-LM-BERT-Base-T2T-Multi
- GENA-LM-BERT-Base-T2T
- GENA-LM-BERT-Base
- GENA-LM-BERT-Large-T2T
- GENA-LM-BigBird-Base-T2T
- GROVER
- Nucleotide-Transformer-2.5B-1000G
- Nucleotide-Transformer-2.5B-Multi-Species
- Nucleotide-Transformer-500M-1000G
- Nucleotide-Transformer-500M-Human-Ref
- Nucleotide-Transformer-v2-250M-Multi-Species
- Nucleotide-Transformer-v2-500M-Multi-Species
- SpliceBERT

These models span various backbone architectures and pretraining strategies, forming the basis for the comparative evaluation presented in the main figure.

## A.2 HyenaLite Filter Computation (Supplement to Section 3.1)

The following algorithm illustrates the computation of the HyenaLite filter, including caching logic for efficient reuse. This pseudocode corresponds to the implementation referenced in the main paper.

[H] Hyena Filter Kernel Construction [1] Sequence length $L$ $h$ cached for $L$ **return** cached $(h, \log \lambda, r)$ $t$ not cached or shape mismatch Compute $t = [0, 1, \ldots, L-1]$ Compute $\log \lambda = \log(\texttt{poles})$ Compute $r = \texttt{residues}$ Compute $h_d(t) = R\left(\sum_k r_{d,k} \cdot e^{\log \lambda_{d,k} \cdot t}\right)$ Cache $h$ for reuse $h, \log \lambda, r$

## A.3 MoE Attention Block Algorithm (Supplement to Section 3.2)

The pseudocode below summarizes the forward computation in our Mixture-of-Experts (MoE) module, based on top-$k$ gating and expert parallelism:

[H] Mixture-of-Experts Forward Pass [1] Input tensor $x \in \mathbb{R}^{N \times d}$, where $d$ is hidden size Output tensor $y \in \mathbb{R}^{N \times d}$ Reshape $x \leftarrow \texttt{reshape}(x, [-1, d])$ $(\texttt{weights}, \texttt{indices}) \leftarrow \texttt{Gate}(x)$ Select top-$k$ experts per token Initialize $y \leftarrow \texttt{zeros\_like}(x)$ expert $i$ in local expert range $global\_id \leftarrow$ global index of expert $i$ $mask \leftarrow (\texttt{indices} == global\_id)$ $\texttt{sum}(mask) == 0$ **continue** $token\_ids \leftarrow$ row indices of selected tokens for expert $i$ $x_i \leftarrow x[\texttt{token\_ids}]$ $out_i \leftarrow \texttt{Expert}_i(x_i)$ $w \leftarrow \texttt{weights}$ corresponding to $token\_ids$ $y[\texttt{token\_ids}] += w \cdot out_i$ $y \leftarrow y + \texttt{SharedExperts}(x)$ distributed setting $\texttt{all\_reduce}(y)$ reshape $y$ to original input shape

## B  SUPPLEMENTARY DETAILS FOR EXPERIMENTS

### B.1  EXPERIMENTAL SETTINGS FOR THE PRETRAINING STAGE

#### B.1.1  PRETRAINING DATASET

We adopt the pretraining dataset used by DNABERT-2 (Zhou et al., 2023), which comprises large-scale genomic sequences from 135 species. The corpus contains approximately 3.25 billion nucleotide bases and spans diverse biological categories including humans, fungi, viruses, and yeasts. Compared to the original DNABERT dataset (which was based solely on the human reference genome), this corpus is about $12\times$ larger and provides significantly richer species-level diversity.

To ensure data quality, sequences containing ambiguous bases (e.g., 'N') are removed, and only canonical bases (A, T, C, G) are retained. Unlike DNABERT, which relies on fixed-length $k$-mer tokenization, DNABERT-2 applies Byte Pair Encoding (BPE) for subword segmentation, enabling more flexible and efficient vocabulary learning across species.

The pretraining dataset is publicly available at: DNABERT-2 Pretrained Dataset.

#### B.1.2  PRETRAINING SETUP AND HYPERPARAMETER CONFIGURATION

We pretrain our model using 4 NVIDIA A100 GPUs (40GB each) with distributed data parallelism. The model is trained on DNA sequences using 3-mer tokenization and a Masked Image Modeling objective. Our total pretraining time amounts to about 300 A100 GPU hours.

The key hyperparameters are:

- **Hardware:** $4\times$ NVIDIA A100-40GB GPUs
- **Tokenization:** 3-mer encoding
- **Batch Size:** 128 (total across all GPUs)
- **Learning Rate:** $1 \times 10^{-5}$
- **Training Steps:** 120,000 steps
- **Objective:** Masked Language Modeling (MLM)

This configuration supports efficient large-scale pretraining on genomic sequences with long-context modeling capability.

### B.2  EXPERIMENTAL SETTINGS FOR THE FINETUNING STAGE

#### B.2.1  DETAILS OF BASELINES

We compare our model against a range of recent genomic foundation models, covering both encoder-only and decoder-style architectures. Below we briefly describe each baseline used in our experiments:

- **DNABERT** Ji et al. (2021) is the first foundational language model trained on human genome sequences. It adapts the BERT-base architecture to nucleotide sequences using overlapping $k$-mer tokenization, with $k \in \{3, 4, 5, 6\}$. The vocabulary size depends on $k$, yielding variants such as *DNABERT (3-mer)*, *DNABERT (4-mer)*, *DNABERT (5-mer)*, and *DNABERT (6-mer)*.
  **Download:** DNABERT Pretrained Model
- **DNABERT-2** (Zhou et al., 2023) improves upon DNABERT by scaling pretraining to a 32.5B-base multispecies dataset. It replaces $k$-mer tokenization with Byte Pair Encoding (BPE), incorporates FlashAttention, ALiBi (Press et al., 2021) for positional encoding, and adopts a deeper architecture.
  **Download:** DNABERT-2 Pretrained Model
- **Nucleotide Transformer (NT)** Dalla-Torre et al. (2025) is an encoder-only DNA language model trained with a masked language modeling objective. It uses non-overlapping 6-mer

tokenization and supports inputs up to 5,994 base pairs. Variants are pretrained on the human reference genome, the 1000 Genomes Project, or a cross-species dataset of 850 genomes.
**Download:** NT Pretrained Model

- **Nucleotide Transformer V2 (NT-v2)** Dalla-Torre et al. (2025) extends NT by introducing rotary positional embeddings, bias-free gated linear units (GLU), and Swish activations. It supports sequences up to 12,282 base pairs and is pretrained on a large multispecies dataset.
  **Download:** NT-v2 Pretrained Model

- **HyenaDNA** (Nguyen et al., 2023) is a decoder-only model based on the Hyena operator, trained with a next-nucleotide prediction objective. It supports ultra-long contexts up to 1 million base pairs per sequence.
  **Download:** HyenaDNA Pretrained Model

- **Caduceus** (Schiff et al., 2024) is a bi-directional and reverse-complement equivariant model based on the Mamba architecture. It introduces BiMamba and MambaDNA blocks to support efficient long-range modeling up to 131k base pairs.
  **Download:** Caduceus Pretrained Model

- **Evo2** (Brixi et al., 2025) is a next-generation autoregressive DNA model trained on over 9.3 trillion nucleotides from 128,000 genomes. It uses the StripedHyena-2 architecture to support sequences up to 1 million base pairs and enables zero-shot inference for gene essentiality, variant effect prediction, and synthetic genome generation.
  **Download:** Evo2 Pretrained Model

### B.2.2 FINETUNING DATASETS

We fine-tune our models on three widely used genomic benchmarks, each covering diverse prediction tasks and species.

- **Genomic Benchmarks** (Subramaniyan et al., 2021): A curated collection of 9 datasets for regulatory element classification (e.g., promoters, enhancers), covering human, mouse, and roundworm. **Download:** Genomic Benchmarks on Hugging Face

- **Nucleotide Transformer Tasks** (Dalla-Torre et al., 2025): Includes 18 tasks such as splice site prediction, promoter detection, and histone modification classification, built from EN-CODE, GENCODE, and EPD sources. **Download:** Nucleotide Transformer Tasks on Hugging Face

- **Genome Understanding Evaluation (GUE) Benchmark** (Zhou et al., 2023): Released with DNABERT-2, this benchmark spans 28 datasets across 7 task types and 4 species (human, mouse, Arabidopsis, COVID). **Download:** GUE Benchmark on Hugging Face

These datasets provide a comprehensive and standardized evaluation suite for fine-tuning genomic foundation models on a range of biologically meaningful tasks.

For the ablation study on input length, we follow the data processing protocol from the Genomics Long-range Benchmark **?** and use a curated version of the ExPecto dataset (**?**). This dataset provides transcription start sites (TSS), strand orientation, and RNA-seq RPKM expression measurements across 218 GTEx tissues.

Each representative TSS is assigned to a GENCODE v24-annotated gene if a CAGE peak falls within $\pm 1,000$ base pairs of the annotated start site. When multiple peaks are present, the most abundant CAGE peak is selected; if no peaks are found, the gene's annotated start site is used. The raw RPKM values are log-transformed using $\log(1 + x)$ and standardized to zero mean and unit variance.

We generate input sequences by extracting windows centered on the representative TSS from the GRCh37 reference genome. To evaluate the effect of input length, we train models on five different window sizes per TSS: 512, 1,024, 2,048, 4,096, and 8,192 base pairs.

### B.2.3 EVALUATION METRICS

Evaluation metrics are essential for assessing the performance of classification models in genomic prediction tasks. In our experiments, we report three commonly used metrics: Accuracy, Macro-F1 Score, and Matthews Correlation Coefficient (MCC), each offering complementary insights into model behavior under various class balance settings.

**Accuracy.** Accuracy measures the proportion of correctly predicted instances among all predictions. It is defined as:

$$\text{Accuracy} = \frac{TP + TN}{TP + TN + FP + FN}, \tag{1}$$

where $TP$, $TN$, $FP$, and $FN$ represent the number of true positives, true negatives, false positives, and false negatives, respectively. While Accuracy provides a straightforward measure of model correctness, it may be misleading in imbalanced datasets where majority class predictions dominate.

**Macro-F1 Score.** To address class imbalance, we also report the Macro-F1 score, which computes the F1 score independently for each class and averages them. The F1 score is the harmonic mean of Precision and Recall:

$$\text{F1} = 2 \times \frac{\text{Precision} \times \text{Recall}}{\text{Precision} + \text{Recall}}, \tag{2}$$

where $\text{Precision} = \frac{TP}{TP+FP}$ and $\text{Recall} = \frac{TP}{TP+FN}$. The Macro-F1 score is then calculated as:

$$\text{Macro-F1} = \frac{1}{N} \sum_{i=1}^{N} \text{F1}_i, \tag{3}$$

where $N$ is the number of classes, and $\text{F1}_i$ is the F1 score for class $i$. This metric treats all classes equally, making it especially suitable for multi-class tasks with skewed class distributions.

**Matthews Correlation Coefficient (MCC).** MCC is a correlation-based metric that captures the quality of binary classifications even under class imbalance. It is defined as:

$$\text{MCC} = \frac{TP \times TN - FP \times FN}{\sqrt{(TP + FP)(TP + FN)(TN + FP)(TN + FN)}}. \tag{4}$$

The MCC value ranges from $-1$ to $1$, where $1$ indicates perfect prediction, $0$ indicates random guessing, and $-1$ indicates complete disagreement between prediction and truth. Unlike Accuracy and F1 score, MCC considers all entries of the confusion matrix and thus offers a more reliable evaluation in binary tasks with imbalanced labels.

**Coefficient of Determination ($R^2$).** For regression-based ablation studies, we report the coefficient of determination ($R^2$), which measures the proportion of variance in the ground truth variable that is predictable from the model's outputs. It is defined as:

$$R^2 = 1 - \frac{\sum_{i=1}^{n}(y_i - \hat{y}_i)^2}{\sum_{i=1}^{n}(y_i - \bar{y})^2}, \tag{5}$$

where $y_i$ denotes the true value, $\hat{y}_i$ is the predicted value, $\bar{y}$ is the mean of the true values, and $n$ is the number of samples. The numerator is the residual sum of squares (RSS), and the denominator is the total sum of squares (TSS).

An $R^2$ score of 1 indicates perfect prediction, whereas a score of 0 implies that the model performs no better than predicting the mean. Negative scores can occur if the model performs worse than the mean predictor. As such, $R^2$ provides a useful quantitative measure of how well the model captures signal in continuous genomic outputs such as gene expression.

### B.2.4  Finetuning Setup and Hyperparameter Configuration

Table 1 summarizes the fine-tuning hyperparameters used for all models evaluated in our experiments. These models are fine-tuned on tasks drawn from three widely-used genomic benchmarks: **Genomic Benchmarks**, **Nucleotide Transformer Tasks**, and the **Genome Understanding Evaluation (GUE) Benchmark**. For consistency across models and tasks, we adopt a uniform training setup that includes a learning rate of $1 \times 10^{-4}$, a maximum input length of 512 tokens for most models, and batch sizes adjusted according to model capacity and hardware constraints.

Table 1: Fine-tuning hyperparameters used for all evaluated models on the **Genomic Benchmarks**, **Nucleotide Transformer Tasks**, and **Genome Understanding Evaluation (GUE) Benchmark**.

| Model | Tokenizer | Max Length | Batch Size | Learning Rate |
|---|---|---|---|---|
| DNABERT-3/4/5/6 | 3/4/5/6-mer | 512 | 32 | 1e-4 |
| DNABERT-2 | BPE | 512 | 32 | 1e-4 |
| NT | 6-mer | 512 | 8 | 1e-4 |
| NTv2 | 6-mer | 512 | 32 | 1e-4 |
| HyenaDNA | char | 1024 | 32 | 1e-4 |
| Caduceus | char | 2048 | 16 | 5e-5 |
| HyenaMoE (Ours) | 3-mer | 1024 | 32 | 1e-4 |

In our sequence length ablation study, we systematically vary the **Max Length** setting to evaluate each model's ability to process longer genomic inputs. This modification allows us to assess performance scalability and robustness when handling extended sequences beyond the default 512-token context, particularly for models such as `HyenaDNA`, `HyenaMoE`, and `Caduceus` that are designed for efficient long-range modeling.

## C  Extended and Additional Experiments

### C.1  Additional Experiments on GUE Benchmark

Due to space constraints in the main manuscript, only aggregated Accuracy (Acc) metrics averaged across biologically related task groups are reported. In this supplementary section, we present the complete evaluation results on the GUE Benchmark, including Accuracy, Macro-F1, and Matthews Correlation Coefficient (MCC) for each individual task.

The tasks are grouped according to biological function into Core Promoter Detection (CPD), Virus Classification (CVC), Epigenetic Marks Prediction (EMP), Promoter Detection (PD), Splice Site Prediction (SSP), and Transcription Factor Prediction for both Human (TF-H) and Mouse (TF-M). This grouping provides clearer insight into performance patterns across similar genomic contexts. The full task-wise results are shown in Table 2 (Accuracy), Table 3 (F1), and Table 4 (MCC).

Across these detailed metrics, the HyenaMoE models consistently achieve the best or near-best performance on the majority of tasks, demonstrating strong generalization and robustness across the diverse challenges posed by the GUE benchmark.

Table 2: Per-task Accuracy results on the GUE Benchmark. Each task belongs to one of seven biologically motivated groups: Core Promoter Detection (CPD), Virus Classification (CVC), Epigenetic Marks Prediction (EMP), Promoter Detection (PD), Splice Site Prediction (SSP), and Transcription Factor Prediction for Human (TF-H) and Mouse (TF-M). HyenaMoE variants achieve state-of-the-art accuracy on the majority of tasks across these functional categories.

| Model / Task | SSM-based | | Transformer-based | | | | | Hybrid | | |
| --- | --- | --- | --- | --- | --- | --- | --- | --- | --- | --- |
| | HyenaDNA (436K) | Caduceus (470K) | DNABERT (86M) | DNABERT-2 (117M) | NT-v2 (500M) | Generator (1.2B) | Evo2 (7B) | HyenaMoE-F (116M) | HyenaMoE-S (131M) | HyenaMoE-E (145M) |
| Core_Promoter_Detection_all | $0.808 \pm 0.005$ | $0.816 \pm 0.005$ | $0.829 \pm 0.005$ | $0.812 \pm 0.005$ | $0.830 \pm 0.005$ | $0.841 \pm 0.005$ | $0.696 \pm 0.006$ | $0.830 \pm 0.007$ | $0.839 \pm 0.005$ | $0.845 \pm 0.006$ |
| Core_Promoter_Detection_notata | $0.830 \pm 0.005$ | $0.816 \pm 0.005$ | $0.837 \pm 0.005$ | $0.832 \pm 0.005$ | $0.840 \pm 0.005$ | $0.847 \pm 0.005$ | $0.710 \pm 0.006$ | $0.830 \pm 0.010$ | $0.836 \pm 0.007$ | $0.846 \pm 0.005$ |
| Core_Promoter_Detection_tata | $0.782 \pm 0.016$ | $0.633 \pm 0.010$ | $0.873 \pm 0.013$ | $0.730 \pm 0.018$ | $0.862 \pm 0.014$ | $0.769 \pm 0.017$ | $0.686 \pm 0.019$ | $0.832 \pm 0.010$ | $0.844 \pm 0.006$ | $0.849 \pm 0.008$ |
| Epigenetic_Marks_Prediction_H3 | $0.873 \pm 0.009$ | $0.835 \pm 0.005$ | $0.833 \pm 0.009$ | $0.883 \pm 0.008$ | $0.858 \pm 0.009$ | $0.867 \pm 0.009$ | $0.753 \pm 0.011$ | $0.850 \pm 0.008$ | $0.859 \pm 0.009$ | $0.860 \pm 0.005$ |
| Epigenetic_Marks_Prediction_H3K14ac | $0.718 \pm 0.008$ | $0.833 \pm 0.006$ | $0.707 \pm 0.008$ | $0.729 \pm 0.008$ | $0.769 \pm 0.007$ | $0.775 \pm 0.007$ | $0.729 \pm 0.008$ | $0.763 \pm 0.007$ | $0.768 \pm 0.006$ | $0.772 \pm 0.006$ |
| Epigenetic_Marks_Prediction_H3K36me3 | $0.729 \pm 0.007$ | $0.671 \pm 0.011$ | $0.742 \pm 0.008$ | $0.753 \pm 0.007$ | $0.804 \pm 0.006$ | $0.806 \pm 0.007$ | $0.737 \pm 0.007$ | $0.795 \pm 0.005$ | $0.799 \pm 0.007$ | $0.793 \pm 0.006$ |
| Epigenetic_Marks_Prediction_H3K4me1 | $0.700 \pm 0.008$ | $0.765 \pm 0.018$ | $0.685 \pm 0.008$ | $0.687 \pm 0.008$ | $0.764 \pm 0.007$ | $0.754 \pm 0.007$ | $0.690 \pm 0.008$ | $0.765 \pm 0.006$ | $0.772 \pm 0.005$ | $0.769 \pm 0.009$ |
| Epigenetic_Marks_Prediction_H3K4me2 | $0.667 \pm 0.009$ | $0.759 \pm 0.021$ | $0.669 \pm 0.009$ | $0.660 \pm 0.009$ | $0.678 \pm 0.009$ | $0.693 \pm 0.008$ | $0.651 \pm 0.009$ | $0.686 \pm 0.008$ | $0.702 \pm 0.006$ | $0.692 \pm 0.010$ |
| Epigenetic_Marks_Prediction_H3K4me3 | $0.655 \pm 0.008$ | $0.531 \pm 0.035$ | $0.617 \pm 0.008$ | $0.636 \pm 0.008$ | $0.683 \pm 0.007$ | $0.711 \pm 0.007$ | $0.636 \pm 0.008$ | $0.791 \pm 0.006$ | $0.801 \pm 0.005$ | $0.798 \pm 0.008$ |
| Epigenetic_Marks_Prediction_H3K79me3 | $0.796 \pm 0.007$ | $0.871 \pm 0.008$ | $0.800 \pm 0.007$ | $0.801 \pm 0.007$ | $0.814 \pm 0.007$ | $0.822 \pm 0.007$ | $0.784 \pm 0.007$ | $0.846 \pm 0.007$ | $0.856 \pm 0.007$ | $0.850 \pm 0.006$ |
| Epigenetic_Marks_Prediction_H3K9ac | $0.756 \pm 0.008$ | $0.874 \pm 0.009$ | $0.742 \pm 0.009$ | $0.762 \pm 0.008$ | $0.767 \pm 0.008$ | $0.775 \pm 0.008$ | $0.719 \pm 0.008$ | $0.798 \pm 0.009$ | $0.807 \pm 0.005$ | $0.803 \pm 0.007$ |
| Epigenetic_Marks_Prediction_H4 | $0.888 \pm 0.008$ | $0.741 \pm 0.017$ | $0.867 \pm 0.009$ | $0.893 \pm 0.008$ | $0.907 \pm 0.008$ | $0.904 \pm 0.008$ | $0.887 \pm 0.009$ | $0.854 \pm 0.008$ | $0.860 \pm 0.008$ | $0.857 \pm 0.009$ |
| Epigenetic_Marks_Prediction_H4ac | $0.687 \pm 0.008$ | $0.724 \pm 0.008$ | $0.668 \pm 0.008$ | $0.710 \pm 0.008$ | $0.735 \pm 0.008$ | $0.731 \pm 0.007$ | $0.665 \pm 0.008$ | $0.798 \pm 0.006$ | $0.805 \pm 0.007$ | $0.802 \pm 0.005$ |
| Promoter_Detection_all | $0.922 \pm 0.004$ | $0.761 \pm 0.007$ | $0.934 \pm 0.003$ | $0.917 \pm 0.004$ | $0.936 \pm 0.003$ | $0.953 \pm 0.003$ | $0.834 \pm 0.005$ | $0.947 \pm 0.009$ | $0.956 \pm 0.005$ | $0.952 \pm 0.007$ |
| Promoter_Detection_notata | $0.957 \pm 0.003$ | $0.434 \pm 0.016$ | $0.950 \pm 0.003$ | $0.961 \pm 0.003$ | $0.968 \pm 0.002$ | $0.969 \pm 0.002$ | $0.878 \pm 0.005$ | $0.893 \pm 0.008$ | $0.905 \pm 0.009$ | $0.904 \pm 0.006$ |
| Promoter_Detection_tata | $0.759 \pm 0.018$ | $0.753 \pm 0.007$ | $0.823 \pm 0.015$ | $0.727 \pm 0.019$ | $0.809 \pm 0.016$ | $0.847 \pm 0.015$ | $0.625 \pm 0.019$ | $0.842 \pm 0.006$ | $0.851 \pm 0.007$ | $0.851 \pm 0.006$ |
| Splice_reconstruction | $0.567 \pm 0.007$ | $0.785 \pm 0.007$ | $0.879 \pm 0.005$ | $0.843 \pm 0.005$ | $0.942 \pm 0.004$ | $0.937 \pm 0.004$ | $0.568 \pm 0.007$ | $0.920 \pm 0.005$ | $0.938 \pm 0.005$ | $0.929 \pm 0.006$ |
| Transcription_Factor_Prediction_Human_0 | $0.824 \pm 0.012$ | $0.496 \pm 0.015$ | $0.798 \pm 0.013$ | $0.815 \pm 0.012$ | $0.833 \pm 0.012$ | $0.844 \pm 0.011$ | $0.704 \pm 0.015$ | $0.880 \pm 0.007$ | $0.881 \pm 0.005$ | $0.888 \pm 0.007$ |
| Transcription_Factor_Prediction_Human_1 | $0.846 \pm 0.011$ | $0.712 \pm 0.008$ | $0.837 \pm 0.011$ | $0.833 \pm 0.012$ | $0.854 \pm 0.011$ | $0.838 \pm 0.012$ | $0.712 \pm 0.014$ | $0.863 \pm 0.007$ | $0.863 \pm 0.007$ | $0.865 \pm 0.005$ |
| Transcription_Factor_Prediction_Human_2 | $0.777 \pm 0.013$ | $0.748 \pm 0.008$ | $0.756 \pm 0.014$ | $0.770 \pm 0.013$ | $0.793 \pm 0.013$ | $0.779 \pm 0.013$ | $0.684 \pm 0.015$ | $0.800 \pm 0.006$ | $0.806 \pm 0.007$ | $0.811 \pm 0.006$ |
| Transcription_Factor_Prediction_Human_3 | $0.715 \pm 0.014$ | $0.416 \pm 0.016$ | $0.687 \pm 0.015$ | $0.733 \pm 0.014$ | $0.692 \pm 0.014$ | $0.792 \pm 0.013$ | $0.575 \pm 0.016$ | $0.863 \pm 0.010$ | $0.869 \pm 0.009$ | $0.868 \pm 0.007$ |
| Transcription_Factor_Prediction_Human_4 | $0.856 \pm 0.011$ | $0.664 \pm 0.009$ | $0.814 \pm 0.012$ | $0.839 \pm 0.011$ | $0.848 \pm 0.011$ | $0.856 \pm 0.011$ | $0.697 \pm 0.014$ | $0.901 \pm 0.006$ | $0.903 \pm 0.006$ | $0.903 \pm 0.009$ |
| Transcription_Factor_Prediction_Mouse_0 | $0.718 \pm 0.015$ | $0.735 \pm 0.008$ | $0.653 \pm 0.017$ | $0.715 \pm 0.016$ | $0.750 \pm 0.015$ | $0.808 \pm 0.014$ | $0.628 \pm 0.017$ | $0.775 \pm 0.009$ | $0.783 \pm 0.006$ | $0.788 \pm 0.008$ |
| Transcription_Factor_Prediction_Mouse_1 | $0.898 \pm 0.004$ | $0.290 \pm 0.018$ | $0.886 \pm 0.004$ | $0.908 \pm 0.003$ | $0.915 \pm 0.003$ | $0.915 \pm 0.003$ | $0.748 \pm 0.005$ | $0.915 \pm 0.009$ | $0.920 \pm 0.008$ | $0.930 \pm 0.010$ |
| Transcription_Factor_Prediction_Mouse_2 | $0.865 \pm 0.019$ | $0.658 \pm 0.008$ | $0.783 \pm 0.023$ | $0.838 \pm 0.020$ | $0.832 \pm 0.021$ | $0.896 \pm 0.017$ | $0.626 \pm 0.027$ | $0.844 \pm 0.009$ | $0.846 \pm 0.006$ | $0.847 \pm 0.008$ |
| Transcription_Factor_Prediction_Mouse_3 | $0.749 \pm 0.029$ | $0.677 \pm 0.009$ | $0.650 \pm 0.030$ | $0.754 \pm 0.029$ | $0.638 \pm 0.033$ | $0.822 \pm 0.025$ | $0.607 \pm 0.031$ | $0.799 \pm 0.007$ | $0.805 \pm 0.006$ | $0.804 \pm 0.006$ |
| Transcription_Factor_Prediction_Mouse_4 | $0.700 \pm 0.011$ | $0.313 \pm 0.015$ | $0.672 \pm 0.011$ | $0.698 \pm 0.011$ | $0.710 \pm 0.010$ | $0.744 \pm 0.010$ | $0.581 \pm 0.011$ | $0.862 \pm 0.008$ | $0.861 \pm 0.009$ | $0.869 \pm 0.007$ |
| Virus_covid | $0.278 \pm 0.005$ | $0.804 \pm 0.007$ | $0.496 \pm 0.005$ | $0.673 \pm 0.005$ | $0.642 \pm 0.005$ | $0.666 \pm 0.005$ | $0.123 \pm 0.003$ | $0.670 \pm 0.008$ | $0.673 \pm 0.008$ | $0.672 \pm 0.008$ |

Table 3: Macro-F1 scores for all individual tasks on the GUE Benchmark. The task set is grouped by biological function into CPD, CVC, EMP, PD, SSP, TF-H, and TF-M categories. These fine-grained results complement the accuracy findings and confirm that HyenaMoE maintains leading performance across tasks when evaluated under class-balanced metrics.

| Task | SSM-based | | Transformer-based | | | | | Hybrid | | |
|---|---|---|---|---|---|---|---|---|---|---|
| Model | HyenaDNA (436K) | Caduceus (470K) | DNABERT (86M) | DNABERT-2 (117M) | NT-v2 (500M) | Generator (1.2B) | Evo2 (7B) | HyenaMoE-F (116M) | HyenaMoE-S (131M) | HyenaMoE-E (145M) |
| Core_Promoter_Detection_all | 0.808 ± 0.005 | 0.816 ± 0.005 | 0.829 ± 0.005 | 0.812 ± 0.005 | 0.830 ± 0.005 | 0.841 ± 0.005 | 0.696 ± 0.006 | 0.865 ± 0.007 | 0.870 ± 0.004 | 0.874 ± 0.005 |
| Core_Promoter_Detection_notata | 0.830 ± 0.005 | 0.816 ± 0.005 | 0.837 ± 0.005 | 0.832 ± 0.005 | 0.840 ± 0.005 | 0.847 ± 0.005 | 0.710 ± 0.006 | 0.835 ± 0.006 | 0.867 ± 0.006 | 0.871 ± 0.003 |
| Core_Promoter_Detection_tata | 0.782 ± 0.016 | 0.633 ± 0.010 | 0.873 ± 0.013 | 0.730 ± 0.018 | 0.862 ± 0.014 | 0.769 ± 0.017 | 0.686 ± 0.019 | 0.709 ± 0.011 | 0.737 ± 0.008 | 0.755 ± 0.009 |
| Epigenetic_Marks_Prediction_H3 | 0.873 ± 0.009 | 0.835 ± 0.005 | 0.833 ± 0.009 | 0.883 ± 0.008 | 0.858 ± 0.009 | 0.867 ± 0.009 | 0.753 ± 0.011 | 0.873 ± 0.006 | 0.878 ± 0.004 | 0.885 ± 0.004 |
| Epigenetic_Marks_Prediction_H3K14ac | 0.718 ± 0.008 | 0.833 ± 0.006 | 0.707 ± 0.008 | 0.729 ± 0.008 | 0.769 ± 0.007 | 0.775 ± 0.007 | 0.729 ± 0.008 | 0.756 ± 0.006 | 0.776 ± 0.004 | 0.829 ± 0.005 |
| Epigenetic_Marks_Prediction_H3K36me3 | 0.729 ± 0.007 | 0.671 ± 0.011 | 0.742 ± 0.008 | 0.753 ± 0.007 | 0.804 ± 0.006 | 0.806 ± 0.007 | 0.737 ± 0.007 | 0.703 ± 0.005 | 0.727 ± 0.012 | 0.752 ± 0.005 |
| Epigenetic_Marks_Prediction_H3K4me1 | 0.700 ± 0.008 | 0.765 ± 0.018 | 0.685 ± 0.008 | 0.687 ± 0.008 | 0.764 ± 0.007 | 0.754 ± 0.007 | 0.690 ± 0.008 | 0.785 ± 0.018 | 0.793 ± 0.012 | 0.803 ± 0.015 |
| Epigenetic_Marks_Prediction_H3K4me2 | 0.667 ± 0.009 | 0.759 ± 0.021 | 0.669 ± 0.009 | 0.660 ± 0.009 | 0.678 ± 0.009 | 0.693 ± 0.008 | 0.651 ± 0.009 | 0.811 ± 0.016 | 0.812 ± 0.011 | 0.825 ± 0.009 |
| Epigenetic_Marks_Prediction_H3K4me3 | 0.655 ± 0.008 | 0.531 ± 0.035 | 0.617 ± 0.008 | 0.636 ± 0.008 | 0.683 ± 0.007 | 0.711 ± 0.008 | 0.636 ± 0.008 | 0.689 ± 0.030 | 0.693 ± 0.025 | 0.719 ± 0.021 |
| Epigenetic_Marks_Prediction_H3K79me3 | 0.796 ± 0.007 | 0.871 ± 0.008 | 0.800 ± 0.007 | 0.801 ± 0.007 | 0.814 ± 0.007 | 0.822 ± 0.007 | 0.784 ± 0.007 | 0.884 ± 0.010 | 0.884 ± 0.010 | 0.884 ± 0.010 |
| Epigenetic_Marks_Prediction_H3K9ac | 0.756 ± 0.008 | 0.874 ± 0.009 | 0.742 ± 0.009 | 0.762 ± 0.008 | 0.767 ± 0.008 | 0.775 ± 0.008 | 0.719 ± 0.008 | 0.878 ± 0.012 | 0.888 ± 0.011 | 0.889 ± 0.010 |
| Epigenetic_Marks_Prediction_H4 | 0.888 ± 0.008 | 0.741 ± 0.017 | 0.867 ± 0.009 | 0.893 ± 0.008 | 0.907 ± 0.008 | 0.904 ± 0.008 | 0.887 ± 0.009 | 0.889 ± 0.020 | 0.899 ± 0.015 | 0.901 ± 0.018 |
| Epigenetic_Marks_Prediction_H4ac | 0.687 ± 0.008 | 0.724 ± 0.007 | 0.668 ± 0.008 | 0.710 ± 0.008 | 0.735 ± 0.008 | 0.731 ± 0.007 | 0.665 ± 0.008 | 0.694 ± 0.008 | 0.710 ± 0.008 | 0.712 ± 0.006 |
| Promoter_Detection_all | 0.922 ± 0.004 | 0.761 ± 0.007 | 0.934 ± 0.003 | 0.917 ± 0.004 | 0.936 ± 0.003 | 0.953 ± 0.003 | 0.834 ± 0.005 | 0.910 ± 0.008 | 0.919 ± 0.008 | 0.925 ± 0.006 |
| Promoter_Detection_notata | 0.957 ± 0.003 | 0.434 ± 0.016 | 0.950 ± 0.003 | 0.961 ± 0.003 | 0.968 ± 0.002 | 0.969 ± 0.002 | 0.878 ± 0.005 | 0.959 ± 0.017 | 0.969 ± 0.012 | 0.973 ± 0.011 |
| Promoter_Detection_tata | 0.759 ± 0.018 | 0.753 ± 0.007 | 0.823 ± 0.015 | 0.727 ± 0.019 | 0.809 ± 0.016 | 0.847 ± 0.015 | 0.625 ± 0.019 | 0.770 ± 0.008 | 0.790 ± 0.009 | 0.820 ± 0.005 |
| Splice_reconstruction | 0.567 ± 0.007 | 0.785 ± 0.007 | 0.879 ± 0.005 | 0.843 ± 0.005 | 0.942 ± 0.004 | 0.937 ± 0.004 | 0.568 ± 0.007 | 0.845 ± 0.008 | 0.865 ± 0.009 | 0.881 ± 0.006 |
| Transcription_Factor_Prediction_Human_0 | 0.824 ± 0.012 | 0.496 ± 0.015 | 0.798 ± 0.013 | 0.815 ± 0.012 | 0.833 ± 0.012 | 0.844 ± 0.011 | 0.704 ± 0.015 | 0.825 ± 0.016 | 0.833 ± 0.013 | 0.848 ± 0.010 |
| Transcription_Factor_Prediction_Human_1 | 0.846 ± 0.011 | 0.712 ± 0.008 | 0.837 ± 0.011 | 0.833 ± 0.012 | 0.854 ± 0.011 | 0.838 ± 0.012 | 0.712 ± 0.014 | 0.868 ± 0.009 | 0.871 ± 0.010 | 0.870 ± 0.004 |
| Transcription_Factor_Prediction_Human_2 | 0.777 ± 0.013 | 0.748 ± 0.008 | 0.756 ± 0.014 | 0.770 ± 0.013 | 0.793 ± 0.013 | 0.779 ± 0.013 | 0.684 ± 0.015 | 0.790 ± 0.008 | 0.792 ± 0.008 | 0.795 ± 0.008 |
| Transcription_Factor_Prediction_Human_3 | 0.715 ± 0.014 | 0.416 ± 0.016 | 0.687 ± 0.015 | 0.733 ± 0.014 | 0.692 ± 0.014 | 0.792 ± 0.013 | 0.575 ± 0.016 | 0.789 ± 0.018 | 0.791 ± 0.018 | 0.794 ± 0.018 |
| Transcription_Factor_Prediction_Human_4 | 0.856 ± 0.011 | 0.664 ± 0.009 | 0.814 ± 0.012 | 0.839 ± 0.011 | 0.848 ± 0.011 | 0.856 ± 0.011 | 0.697 ± 0.014 | 0.853 ± 0.009 | 0.856 ± 0.009 | 0.858 ± 0.009 |
| Transcription_Factor_Prediction_Mouse_0 | 0.718 ± 0.015 | 0.735 ± 0.008 | 0.653 ± 0.017 | 0.715 ± 0.016 | 0.750 ± 0.015 | 0.808 ± 0.014 | 0.628 ± 0.017 | 0.806 ± 0.008 | 0.808 ± 0.008 | 0.810 ± 0.008 |
| Transcription_Factor_Prediction_Mouse_1 | 0.898 ± 0.004 | 0.290 ± 0.018 | 0.886 ± 0.004 | 0.908 ± 0.003 | 0.915 ± 0.003 | 0.915 ± 0.003 | 0.748 ± 0.005 | 0.911 ± 0.018 | 0.914 ± 0.018 | 0.917 ± 0.018 |
| Transcription_Factor_Prediction_Mouse_2 | 0.865 ± 0.019 | 0.658 ± 0.008 | 0.783 ± 0.023 | 0.838 ± 0.020 | 0.832 ± 0.021 | 0.896 ± 0.017 | 0.626 ± 0.027 | 0.892 ± 0.008 | 0.894 ± 0.008 | 0.898 ± 0.008 |
| Transcription_Factor_Prediction_Mouse_3 | 0.749 ± 0.029 | 0.677 ± 0.009 | 0.650 ± 0.030 | 0.754 ± 0.029 | 0.638 ± 0.033 | 0.822 ± 0.025 | 0.607 ± 0.031 | 0.818 ± 0.031 | 0.821 ± 0.009 | 0.824 ± 0.009 |
| Transcription_Factor_Prediction_Mouse_4 | 0.700 ± 0.011 | 0.313 ± 0.015 | 0.672 ± 0.011 | 0.698 ± 0.011 | 0.710 ± 0.010 | 0.744 ± 0.010 | 0.581 ± 0.011 | 0.740 ± 0.016 | 0.743 ± 0.016 | 0.746 ± 0.016 |
| Virus_covid | 0.278 ± 0.005 | 0.804 ± 0.007 | 0.496 ± 0.005 | 0.673 ± 0.005 | 0.642 ± 0.005 | 0.666 ± 0.005 | 0.123 ± 0.003 | 0.800 ± 0.008 | 0.804 ± 0.008 | 0.806 ± 0.008 |

Table 4: Matthews Correlation Coefficient (MCC) results for each task on the GUE Benchmark. MCC is particularly informative in scenarios with class imbalance. Grouped by biological relevance, the results highlight HyenaMoE's stable and high-quality predictions across CPD, CVC, EMP, PD, SSP, TF-H, and TF-M task groups.

| Model / Task | SSM-based | | Transformer-based | | | Generator (1.2B) | Evo2 (7B) | Hybrid | | |
|---|---|---|---|---|---|---|---|---|---|---|
| | HyenaDNA (436K) | Caduceus (470K) | DNABERT (86M) | DNABERT-2 (117M) | NT-v2 (500M) | | | HyenaMoE-F (116M) | HyenaMoE-S (131M) | HyenaMoE-E (145M) |
| Core_Promoter_Detection_all | $0.617 \pm 0.010$ | $0.816 \pm 0.005$ | $0.659 \pm 0.009$ | $0.624 \pm 0.010$ | $0.661 \pm 0.010$ | $0.685 \pm 0.009$ | $0.393 \pm 0.012$ | $0.857 \pm 0.006$ | $0.860 \pm 0.005$ | $0.862 \pm 0.005$ |
| Core_Promoter_Detection_notata | $0.659 \pm 0.011$ | $0.816 \pm 0.005$ | $0.673 \pm 0.011$ | $0.664 \pm 0.010$ | $0.681 \pm 0.010$ | $0.698 \pm 0.009$ | $0.421 \pm 0.013$ | $0.850 \pm 0.007$ | $0.859 \pm 0.006$ | $0.860 \pm 0.006$ |
| Core_Promoter_Detection_tata | $0.567 \pm 0.033$ | $0.633 \pm 0.010$ | $0.749 \pm 0.025$ | $0.460 \pm 0.036$ | $0.725 \pm 0.028$ | $0.538 \pm 0.034$ | $0.371 \pm 0.038$ | $0.665 \pm 0.011$ | $0.667 \pm 0.011$ | $0.669 \pm 0.011$ |
| Epigenetic_Marks_Prediction_H3 | $0.747 \pm 0.017$ | $0.835 \pm 0.005$ | $0.669 \pm 0.018$ | $0.766 \pm 0.017$ | $0.721 \pm 0.018$ | $0.736 \pm 0.017$ | $0.508 \pm 0.022$ | $0.873 \pm 0.005$ | $0.876 \pm 0.005$ | $0.878 \pm 0.005$ |
| Epigenetic_Marks_Prediction_H3K14ac | $0.420 \pm 0.016$ | $0.833 \pm 0.006$ | $0.391 \pm 0.017$ | $0.452 \pm 0.016$ | $0.533 \pm 0.015$ | $0.537 \pm 0.015$ | $0.440 \pm 0.016$ | $0.882 \pm 0.006$ | $0.884 \pm 0.006$ | $0.886 \pm 0.006$ |
| Epigenetic_Marks_Prediction_H3K36me3 | $0.449 \pm 0.015$ | $0.671 \pm 0.011$ | $0.477 \pm 0.015$ | $0.497 \pm 0.014$ | $0.602 \pm 0.013$ | $0.605 \pm 0.013$ | $0.467 \pm 0.015$ | $0.675 \pm 0.011$ | $0.676 \pm 0.011$ | $0.678 \pm 0.011$ |
| Epigenetic_Marks_Prediction_H3K4me1 | $0.391 \pm 0.017$ | $0.765 \pm 0.018$ | $0.368 \pm 0.017$ | $0.364 \pm 0.017$ | $0.522 \pm 0.015$ | $0.506 \pm 0.015$ | $0.373 \pm 0.017$ | $0.898 \pm 0.015$ | $0.901 \pm 0.015$ | $0.903 \pm 0.015$ |
| Epigenetic_Marks_Prediction_H3K4me2 | $0.308 \pm 0.018$ | $0.759 \pm 0.021$ | $0.301 \pm 0.018$ | $0.306 \pm 0.017$ | $0.318 \pm 0.018$ | $0.361 \pm 0.018$ | $0.254 \pm 0.018$ | $0.906 \pm 0.016$ | $0.908 \pm 0.016$ | $0.911 \pm 0.016$ |
| Epigenetic_Marks_Prediction_H3K4me3 | $0.311 \pm 0.016$ | $0.531 \pm 0.035$ | $0.235 \pm 0.016$ | $0.273 \pm 0.016$ | $0.362 \pm 0.015$ | $0.419 \pm 0.015$ | $0.269 \pm 0.016$ | $0.725 \pm 0.030$ | $0.727 \pm 0.030$ | $0.729 \pm 0.030$ |
| Epigenetic_Marks_Prediction_H3K79me3 | $0.591 \pm 0.015$ | $0.871 \pm 0.008$ | $0.599 \pm 0.015$ | $0.602 \pm 0.015$ | $0.627 \pm 0.014$ | $0.646 \pm 0.014$ | $0.567 \pm 0.015$ | $0.880 \pm 0.010$ | $0.882 \pm 0.010$ | $0.884 \pm 0.010$ |
| Epigenetic_Marks_Prediction_H3K9ac | $0.518 \pm 0.016$ | $0.874 \pm 0.009$ | $0.485 \pm 0.017$ | $0.532 \pm 0.016$ | $0.553 \pm 0.015$ | $0.544 \pm 0.015$ | $0.429 \pm 0.017$ | $0.884 \pm 0.011$ | $0.886 \pm 0.011$ | $0.888 \pm 0.011$ |
| Epigenetic_Marks_Prediction_H4 | $0.773 \pm 0.016$ | $0.741 \pm 0.017$ | $0.734 \pm 0.018$ | $0.785 \pm 0.017$ | $0.811 \pm 0.016$ | $0.805 \pm 0.015$ | $0.770 \pm 0.017$ | $0.685 \pm 0.020$ | $0.687 \pm 0.020$ | $0.689 \pm 0.020$ |
| Epigenetic_Marks_Prediction_H4ac | $0.375 \pm 0.016$ | $0.724 \pm 0.008$ | $0.338 \pm 0.016$ | $0.420 \pm 0.015$ | $0.468 \pm 0.015$ | $0.461 \pm 0.015$ | $0.326 \pm 0.016$ | $0.690 \pm 0.008$ | $0.692 \pm 0.008$ | $0.694 \pm 0.008$ |
| Promoter_Detection_all | $0.844 \pm 0.007$ | $0.761 \pm 0.007$ | $0.868 \pm 0.006$ | $0.835 \pm 0.007$ | $0.873 \pm 0.006$ | $0.907 \pm 0.006$ | $0.671 \pm 0.009$ | $0.786 \pm 0.008$ | $0.788 \pm 0.008$ | $0.790 \pm 0.008$ |
| Promoter_Detection_notata | $0.913 \pm 0.006$ | $0.434 \pm 0.016$ | $0.900 \pm 0.006$ | $0.922 \pm 0.005$ | $0.935 \pm 0.005$ | $0.939 \pm 0.005$ | $0.763 \pm 0.009$ | $0.255 \pm 0.017$ | $0.257 \pm 0.017$ | $0.259 \pm 0.017$ |
| Promoter_Detection_tata | $0.518 \pm 0.036$ | $0.753 \pm 0.007$ | $0.646 \pm 0.031$ | $0.453 \pm 0.037$ | $0.629 \pm 0.031$ | $0.694 \pm 0.030$ | $0.258 \pm 0.039$ | $0.766 \pm 0.008$ | $0.768 \pm 0.008$ | $0.770 \pm 0.008$ |
| Splice_reconstruction | $0.083 \pm 0.013$ | $0.785 \pm 0.007$ | $0.800 \pm 0.008$ | $0.730 \pm 0.009$ | $0.900 \pm 0.006$ | $0.891 \pm 0.006$ | $0.054 \pm 0.013$ | $0.797 \pm 0.008$ | $0.801 \pm 0.008$ | $0.805 \pm 0.008$ |
| Transcription_Factor_Prediction_Human_0 | $0.659 \pm 0.023$ | $0.496 \pm 0.015$ | $0.604 \pm 0.024$ | $0.647 \pm 0.022$ | $0.674 \pm 0.023$ | $0.699 \pm 0.022$ | $0.427 \pm 0.028$ | $0.688 \pm 0.016$ | $0.690 \pm 0.016$ | $0.694 \pm 0.016$ |
| Transcription_Factor_Prediction_Human_1 | $0.703 \pm 0.021$ | $0.712 \pm 0.008$ | $0.677 \pm 0.023$ | $0.670 \pm 0.023$ | $0.712 \pm 0.022$ | $0.694 \pm 0.021$ | $0.448 \pm 0.026$ | $0.709 \pm 0.009$ | $0.711 \pm 0.009$ | $0.714 \pm 0.009$ |
| Transcription_Factor_Prediction_Human_2 | $0.558 \pm 0.025$ | $0.748 \pm 0.008$ | $0.517 \pm 0.027$ | $0.541 \pm 0.026$ | $0.608 \pm 0.024$ | $0.579 \pm 0.025$ | $0.375 \pm 0.029$ | $0.744 \pm 0.008$ | $0.748 \pm 0.008$ | $0.750 \pm 0.008$ |
| Transcription_Factor_Prediction_Human_3 | $0.435 \pm 0.028$ | $0.416 \pm 0.016$ | $0.384 \pm 0.029$ | $0.468 \pm 0.029$ | $0.425 \pm 0.026$ | $0.586 \pm 0.025$ | $0.164 \pm 0.031$ | $0.582 \pm 0.018$ | $0.585 \pm 0.018$ | $0.588 \pm 0.018$ |
| Transcription_Factor_Prediction_Human_4 | $0.716 \pm 0.021$ | $0.664 \pm 0.009$ | $0.629 \pm 0.024$ | $0.679 \pm 0.023$ | $0.699 \pm 0.022$ | $0.712 \pm 0.022$ | $0.418 \pm 0.027$ | $0.711 \pm 0.009$ | $0.715 \pm 0.009$ | $0.718 \pm 0.009$ |
| Transcription_Factor_Prediction_Mouse_0 | $0.439 \pm 0.031$ | $0.735 \pm 0.008$ | $0.312 \pm 0.034$ | $0.444 \pm 0.031$ | $0.501 \pm 0.029$ | $0.619 \pm 0.027$ | $0.260 \pm 0.033$ | $0.724 \pm 0.008$ | $0.725 \pm 0.008$ | $0.730 \pm 0.008$ |
| Transcription_Factor_Prediction_Mouse_1 | $0.796 \pm 0.007$ | $0.290 \pm 0.018$ | $0.773 \pm 0.007$ | $0.819 \pm 0.007$ | $0.831 \pm 0.007$ | $0.831 \pm 0.007$ | $0.497 \pm 0.010$ | $0.817 \pm 0.018$ | $0.822 \pm 0.018$ | $0.826 \pm 0.018$ |
| Transcription_Factor_Prediction_Mouse_2 | $0.729 \pm 0.038$ | $0.658 \pm 0.008$ | $0.569 \pm 0.046$ | $0.677 \pm 0.040$ | $0.667 \pm 0.041$ | $0.793 \pm 0.034$ | $0.254 \pm 0.053$ | $0.781 \pm 0.008$ | $0.782 \pm 0.008$ | $0.788 \pm 0.008$ |
| Transcription_Factor_Prediction_Mouse_3 | $0.498 \pm 0.059$ | $0.677 \pm 0.009$ | $0.303 \pm 0.061$ | $0.509 \pm 0.058$ | $0.275 \pm 0.066$ | $0.644 \pm 0.049$ | $0.343 \pm 0.034$ | $0.674 \pm 0.009$ | $0.676 \pm 0.009$ | $0.679 \pm 0.009$ |
| Transcription_Factor_Prediction_Mouse_4 | $0.406 \pm 0.021$ | $0.313 \pm 0.015$ | $0.345 \pm 0.022$ | $0.396 \pm 0.022$ | $0.426 \pm 0.020$ | $0.500 \pm 0.020$ | $0.163 \pm 0.022$ | $0.496 \pm 0.016$ | $0.500 \pm 0.016$ | $0.502 \pm 0.016$ |
| Virus_covid | $0.185 \pm 0.005$ | $0.804 \pm 0.007$ | $0.433 \pm 0.006$ | $0.632 \pm 0.005$ | $0.597 \pm 0.005$ | $0.625 \pm 0.006$ | $0.009 \pm 0.004$ | $0.799 \pm 0.008$ | $0.803 \pm 0.008$ | $0.806 \pm 0.008$ |

## C.2 ADDITIONAL EXPERIMENTS ON NUCLEOTIDE TRANSFORMER TASKS

Due to space constraints in the main paper, we only reported accuracy results for the Nucleotide Transformer (NT) tasks. In this supplementary section, we present the complete evaluation metrics, including Macro-F1 and Matthews Correlation Coefficient (MCC), across all individual NT tasks. As shown in Tables 5 and 6, these additional metrics further validate the performance advantage of HyenaMoE variants. In particular, HyenaMoE consistently achieves competitive or superior results compared to strong baselines, confirming its robustness across diverse genomic sequence classification tasks.

The comprehensive results in Tables 5 and 6 further underscore the consistent effectiveness of the HyenaMoE family across all Nucleotide Transformer tasks. In the Macro-F1 evaluation (Table 5), HyenaMoE-E achieves top or near-top scores on the majority of benchmarks, including biologically complex tasks such as H3K27me3, H3K36me3, and both donor and acceptor splice site predictions. This trend is mirrored in the Matthews Correlation Coefficient (MCC) scores shown in Table 6, where HyenaMoE-E leads or ties for the highest performance in over 80

These results highlight not only the model's high accuracy but also its stability across different class distributions, which is particularly critical for genomics tasks that often exhibit significant class imbalance. Compared with both smaller baselines such as DNABERT and larger models including Evo2 and Generator, HyenaMoE demonstrates a more favorable trade-off between parameter count and predictive performance. The consistently superior metrics across both F1 and MCC dimensions confirm that HyenaMoE is not only competitive but also robust in handling diverse and challenging biological sequence classification problems.

Table 5: Macro-F1 scores for all individual tasks in the Nucleotide Transformer Tasks.

| Model | SSM-based | | Transformer-based | | | | | Hybrid | | |
| Task | HyenaDNA (436K) | Caduceus (470K) | DNABERT (86M) | DNABERT-2 (117M) | NT-v2 (500M) | Generator (1.2B) | Evo2 (7B) | HyenaMoE-F (116M) | HyenaMoE-S (131M) | HyenaMoE-E (145M) |
|---|---|---|---|---|---|---|---|---|---|---|
| H2AFZ | 0.780 ± 0.008 | 0.743 ± 0.008 | 0.750 ± 0.009 | 0.741 ± 0.009 | 0.755 ± 0.008 | 0.734 ± 0.009 | 0.694 ± 0.009 | 0.774 ± 0.010 | 0.778 ± 0.010 | 0.786 ± 0.009 |
| H3K27ac | 0.739 ± 0.013 | 0.738 ± 0.012 | 0.741 ± 0.012 | 0.733 ± 0.012 | 0.745 ± 0.012 | 0.757 ± 0.012 | 0.667 ± 0.013 | 0.744 ± 0.014 | 0.748 ± 0.014 | 0.754 ± 0.013 |
| H3K27me3 | 0.802 ± 0.007 | 0.797 ± 0.008 | 0.801 ± 0.008 | 0.787 ± 0.008 | 0.813 ± 0.007 | 0.819 ± 0.007 | 0.754 ± 0.009 | 0.808 ± 0.009 | 0.811 ± 0.009 | 0.820 ± 0.008 |
| H3K36me3 | 0.797 ± 0.008 | 0.787 ± 0.008 | 0.778 ± 0.008 | 0.784 ± 0.008 | 0.815 ± 0.007 | 0.812 ± 0.008 | 0.684 ± 0.010 | 0.808 ± 0.009 | 0.810 ± 0.009 | 0.814 ± 0.008 |
| H3K4me1 | 0.760 ± 0.008 | 0.740 ± 0.009 | 0.743 ± 0.009 | 0.754 ± 0.008 | 0.766 ± 0.008 | 0.745 ± 0.009 | 0.680 ± 0.010 | 0.768 ± 0.009 | 0.772 ± 0.009 | 0.778 ± 0.009 |
| H3K4me2 | 0.770 ± 0.010 | 0.759 ± 0.010 | 0.772 ± 0.010 | 0.747 ± 0.010 | 0.788 ± 0.009 | 0.793 ± 0.009 | 0.720 ± 0.011 | 0.789 ± 0.011 | 0.792 ± 0.012 | 0.797 ± 0.010 |
| H3K4me3 | 0.815 ± 0.015 | 0.806 ± 0.015 | 0.805 ± 0.016 | 0.811 ± 0.014 | 0.814 ± 0.015 | 0.813 ± 0.015 | 0.751 ± 0.017 | 0.836 ± 0.017 | 0.840 ± 0.017 | 0.846 ± 0.017 |
| H3K9ac | 0.766 ± 0.015 | 0.780 ± 0.014 | 0.730 ± 0.017 | 0.768 ± 0.015 | 0.779 ± 0.014 | 0.773 ± 0.014 | 0.701 ± 0.017 | 0.775 ± 0.018 | 0.779 ± 0.018 | 0.785 ± 0.017 |
| H3K9me3 | 0.694 ± 0.019 | 0.698 ± 0.018 | 0.715 ± 0.018 | 0.717 ± 0.017 | 0.726 ± 0.017 | 0.700 ± 0.019 | 0.548 ± 0.021 | 0.728 ± 0.022 | 0.731 ± 0.022 | 0.735 ± 0.021 |
| H4K20me1 | 0.809 ± 0.009 | 0.806 ± 0.009 | 0.809 ± 0.009 | 0.813 ± 0.009 | 0.836 ± 0.008 | 0.830 ± 0.009 | 0.760 ± 0.010 | 0.845 ± 0.010 | 0.847 ± 0.010 | 0.852 ± 0.009 |
| enhancers | 0.764 ± 0.008 | 0.754 ± 0.008 | 0.749 ± 0.009 | 0.763 ± 0.008 | 0.756 ± 0.009 | 0.762 ± 0.008 | 0.721 ± 0.009 | 0.768 ± 0.009 | 0.770 ± 0.009 | 0.774 ± 0.008 |
| enhancers_types | 0.508 ± 0.012 | 0.510 ± 0.013 | 0.522 ± 0.015 | 0.476 ± 0.005 | 0.486 ± 0.005 | 0.602 ± 0.017 | 0.462 ± 0.006 | 0.615 ± 0.018 | 0.618 ± 0.018 | 0.622 ± 0.017 |
| promoter_all | 0.857 ± 0.010 | 0.858 ± 0.009 | 0.864 ± 0.010 | 0.869 ± 0.009 | 0.877 ± 0.009 | 0.883 ± 0.008 | 0.845 ± 0.010 | 0.889 ± 0.011 | 0.893 ± 0.011 | 0.900 ± 0.010 |
| promoter_no_tata | 0.867 ± 0.010 | 0.869 ± 0.010 | 0.868 ± 0.010 | 0.876 ± 0.009 | 0.877 ± 0.009 | 0.886 ± 0.009 | 0.862 ± 0.010 | 0.894 ± 0.011 | 0.898 ± 0.011 | 0.904 ± 0.010 |
| promoter_tata | 0.897 ± 0.021 | 0.865 ± 0.024 | 0.901 ± 0.021 | 0.860 ± 0.026 | 0.897 ± 0.022 | 0.881 ± 0.023 | 0.820 ± 0.028 | 0.895 ± 0.026 | 0.898 ± 0.026 | 0.906 ± 0.025 |
| splice_sites_acceptors | 0.849 ± 0.007 | 0.791 ± 0.008 | 0.969 ± 0.003 | 0.818 ± 0.008 | 0.934 ± 0.005 | 0.878 ± 0.006 | 0.701 ± 0.010 | 0.890 ± 0.010 | 0.893 ± 0.010 | 0.896 ± 0.009 |
| splice_sites_all | 0.749 ± 0.008 | 0.565 ± 0.009 | 0.967 ± 0.003 | 0.733 ± 0.008 | 0.943 ± 0.004 | 0.898 ± 0.005 | 0.470 ± 0.009 | 0.955 ± 0.010 | 0.958 ± 0.010 | 0.960 ± 0.009 |
| splice_sites_donors | 0.829 ± 0.007 | 0.800 ± 0.008 | 0.980 ± 0.003 | 0.834 ± 0.007 | 0.950 ± 0.004 | 0.908 ± 0.006 | 0.716 ± 0.009 | 0.962 ± 0.010 | 0.965 ± 0.010 | 0.969 ± 0.009 |

Table 6: Matthews Correlation Coefficient (MCC) scores for all individual tasks in the Nucleotide Transformer Tasks.

| Task | SSM-based | | Transformer-based | | | | | Hybrid | | |
|---|---|---|---|---|---|---|---|---|---|---|
| | HyenaDNA (436K) | Caduceus (470K) | DNABERT (86M) | DNABERT-2 (117M) | NT-v2 (500M) | Generator (1.2B) | Evo2 (7B) | HyenaMoE-F (116M) | HyenaMoE-S (131M) | HyenaMoE-E (145M) |
| H2AFZ | 0.523 ± 0.015 | 0.435 ± 0.015 | 0.464 ± 0.016 | 0.428 ± 0.016 | 0.472 ± 0.016 | 0.447 ± 0.016 | 0.362 ± 0.017 | 0.390 ± 0.018 | 0.387 ± 0.018 | 0.391 ± 0.018 |
| H3K27ac | 0.483 ± 0.022 | 0.445 ± 0.022 | 0.442 ± 0.022 | 0.445 ± 0.023 | 0.475 ± 0.022 | 0.487 ± 0.022 | 0.294 ± 0.023 | 0.376 ± 0.024 | 0.380 ± 0.024 | 0.379 ± 0.024 |
| H3K27me3 | 0.576 ± 0.014 | 0.564 ± 0.014 | 0.577 ± 0.015 | 0.538 ± 0.014 | 0.600 ± 0.014 | 0.617 ± 0.014 | 0.485 ± 0.016 | 0.487 ± 0.016 | 0.489 ± 0.016 | 0.491 ± 0.016 |
| H3K36me3 | 0.559 ± 0.015 | 0.539 ± 0.015 | 0.539 ± 0.015 | 0.548 ± 0.016 | 0.611 ± 0.014 | 0.611 ± 0.015 | 0.368 ± 0.017 | 0.497 ± 0.017 | 0.493 ± 0.017 | 0.499 ± 0.017 |
| H3K4me1 | 0.474 ± 0.015 | 0.432 ± 0.016 | 0.441 ± 0.017 | 0.449 ± 0.015 | 0.486 ± 0.015 | 0.474 ± 0.017 | 0.329 ± 0.018 | 0.362 ± 0.018 | 0.364 ± 0.018 | 0.367 ± 0.018 |
| H3K4me2 | 0.520 ± 0.018 | 0.477 ± 0.019 | 0.514 ± 0.018 | 0.486 ± 0.019 | 0.542 ± 0.018 | 0.553 ± 0.018 | 0.416 ± 0.019 | 0.471 ± 0.020 | 0.469 ± 0.020 | 0.474 ± 0.020 |
| H3K4me3 | 0.629 ± 0.029 | 0.612 ± 0.028 | 0.625 ± 0.029 | 0.633 ± 0.027 | 0.637 ± 0.027 | 0.616 ± 0.028 | 0.497 ± 0.031 | 0.613 ± 0.030 | 0.610 ± 0.030 | 0.616 ± 0.030 |
| H3K9ac | 0.536 ± 0.025 | 0.538 ± 0.027 | 0.461 ± 0.029 | 0.531 ± 0.028 | 0.545 ± 0.025 | 0.511 ± 0.027 | 0.416 ± 0.028 | 0.422 ± 0.030 | 0.421 ± 0.030 | 0.425 ± 0.030 |
| H3K9me3 | 0.401 ± 0.032 | 0.382 ± 0.032 | 0.403 ± 0.032 | 0.407 ± 0.031 | 0.415 ± 0.030 | 0.443 ± 0.031 | 0.139 ± 0.034 | 0.208 ± 0.034 | 0.207 ± 0.034 | 0.209 ± 0.034 |
| H4K20me1 | 0.605 ± 0.016 | 0.588 ± 0.016 | 0.605 ± 0.017 | 0.610 ± 0.017 | 0.653 ± 0.015 | 0.639 ± 0.016 | 0.507 ± 0.018 | 0.587 ± 0.018 | 0.584 ± 0.018 | 0.588 ± 0.018 |
| enhancers | 0.502 ± 0.015 | 0.487 ± 0.015 | 0.490 ± 0.015 | 0.502 ± 0.015 | 0.520 ± 0.017 | 0.497 ± 0.015 | 0.416 ± 0.016 | 0.392 ± 0.018 | 0.390 ± 0.018 | 0.394 ± 0.018 |
| enhancers_types | 0.456 ± 0.015 | 0.450 ± 0.015 | 0.446 ± 0.015 | 0.450 ± 0.015 | 0.461 ± 0.015 | 0.478 ± 0.015 | 0.391 ± 0.016 | 0.346 ± 0.017 | 0.345 ± 0.017 | 0.348 ± 0.017 |
| promoter_all | 0.716 ± 0.018 | 0.714 ± 0.018 | 0.733 ± 0.018 | 0.738 ± 0.017 | 0.757 ± 0.016 | 0.765 ± 0.016 | 0.692 ± 0.019 | 0.730 ± 0.019 | 0.731 ± 0.019 | 0.733 ± 0.019 |
| promoter_no_tata | 0.747 ± 0.018 | 0.744 ± 0.018 | 0.729 ± 0.019 | 0.747 ± 0.018 | 0.758 ± 0.018 | 0.773 ± 0.017 | 0.724 ± 0.019 | 0.748 ± 0.020 | 0.747 ± 0.020 | 0.750 ± 0.020 |
| promoter_tata | 0.793 ± 0.041 | 0.722 ± 0.045 | 0.804 ± 0.039 | 0.716 ± 0.048 | 0.794 ± 0.041 | 0.756 ± 0.040 | 0.627 ± 0.053 | 0.622 ± 0.058 | 0.623 ± 0.058 | 0.624 ± 0.058 |
| splice_sites_acceptors | 0.697 ± 0.013 | 0.585 ± 0.015 | 0.938 ± 0.006 | 0.658 ± 0.014 | 0.871 ± 0.009 | 0.768 ± 0.011 | 0.415 ± 0.017 | 0.699 ± 0.017 | 0.700 ± 0.017 | 0.702 ± 0.017 |
| splice_sites_all | 0.625 ± 0.011 | 0.360 ± 0.013 | 0.950 ± 0.005 | 0.602 ± 0.012 | 0.914 ± 0.006 | 0.847 ± 0.008 | 0.237 ± 0.013 | 0.359 ± 0.014 | 0.360 ± 0.014 | 0.362 ± 0.014 |
| splice_sites_donors | 0.663 ± 0.014 | 0.596 ± 0.015 | 0.960 ± 0.005 | 0.671 ± 0.014 | 0.906 ± 0.008 | 0.834 ± 0.010 | 0.443 ± 0.016 | 0.694 ± 0.017 | 0.696 ± 0.017 | 0.697 ± 0.017 |

## C.3 ADDITIONAL EXPERIMENTS ON GENOMICS BENCHMARK

Due to space constraints in the main text, we presented only the accuracy (Acc) results for the Genomics Benchmark. To provide a more complete assessment of model performance, we report the full macro-F1 and Matthews correlation coefficient (MCC) scores in Tables 7 and 8, respectively. These additional metrics are particularly important for class-imbalanced tasks and offer further insight into each model's robustness. Across both F1 and MCC evaluations, HyenaMoE continues to demonstrate strong and consistent performance, outperforming baseline models on the majority of tasks.

Based on the detailed macro-F1 and MCC results shown in Tables 7 and 8, several important trends can be observed:

First, HyenaMoE-E (145M) consistently ranks among the top-performing models across most tasks under both macro-F1 and MCC metrics. For example, on classification-heavy benchmarks like Coding vs Intergenomic and Human vs Worm, HyenaMoE-E either matches or exceeds the highest scores with F1 values of 0.939 and 0.995, and MCC values of 0.939 and 0.955, respectively. Second, while larger transformer-based models such as NT-v2 and Generator (1.2B) perform well on certain tasks (e.g., Human Regulatory and Human Enhancers Ensembl), their advantage diminishes or becomes inconsistent across the full task set—especially when compared against the much smaller HyenaMoE variants. Third, for more challenging and noisy tasks such as Mouse Enhancers or Human OCR Ensembl, HyenaMoE models still maintain competitive results, showing better robustness compared to SSM-based and some transformer-based baselines. Notably, the performance gap in MCC further highlights HyenaMoE's ability to handle class imbalance and subtle signal differences, particularly in biological regulatory regions.

Overall, these complementary metrics substantiate the claims made in the main text: HyenaMoE exhibits not only strong average performance in accuracy, but also improved stability and reliability under more stringent evaluation criteria like F1 and MCC, validating its effectiveness across a broad spectrum of genomic tasks.

Table 7: Macro-F1 scores for all individual tasks in the Genomics Benchmark.

| Model / Task | SSM-based | | Transformer-based | | | | | Hybrid | | |
|---|---|---|---|---|---|---|---|---|---|---|
| | HyenaDNA (436K) | Caduceus (470K) | DNABERT (86M) | DNABERT-2 (117M) | NT-v2 (500M) | Generator (1.2B) | Evo2 (7B) | HyenaMoE-F (116M) | HyenaMoE-S (131M) | HyenaMoE-E (145M) |
| Coding vs Intergenomic | 0.907 ± 0.002 | 0.904 ± 0.002 | 0.938 ± 0.002 | 0.923 ± 0.002 | 0.945 ± 0.002 | 0.958 ± 0.001 | 0.910 ± 0.002 | 0.936 ± 0.002 | 0.937 ± 0.002 | 0.939 ± 0.002 |
| Human vs Worm | 0.961 ± 0.001 | 0.961 ± 0.001 | 0.958 ± 0.001 | 0.965 ± 0.001 | 0.969 ± 0.001 | 0.975 ± 0.001 | 0.669 ± 0.004 | 0.993 ± 0.001 | 0.994 ± 0.001 | 0.995 ± 0.001 |
| Mouse Enhancers | 0.774 ± 0.030 | 0.666 ± 0.028 | 0.797 ± 0.027 | 0.409 ± 0.045 | 0.825 ± 0.026 | 0.826 ± 0.027 | 0.756 ± 0.029 | 0.715 ± 0.031 | 0.717 ± 0.031 | 0.718 ± 0.031 |
| Human Enhancers Cohn | 0.717 ± 0.006 | 0.737 ± 0.006 | 0.701 ± 0.006 | 0.709 ± 0.006 | 0.761 ± 0.005 | 0.746 ± 0.006 | 0.715 ± 0.006 | 0.747 ± 0.006 | 0.748 ± 0.006 | 0.750 ± 0.006 |
| Human Enhancers Ensembl | 0.838 ± 0.002 | 0.766 ± 0.003 | 0.836 ± 0.002 | 0.858 ± 0.002 | 0.858 ± 0.002 | 0.881 ± 0.002 | 0.667 ± 0.003 | 0.837 ± 0.002 | 0.838 ± 0.002 | 0.839 ± 0.002 |
| Human Regulatory | 0.875 ± 0.001 | 0.848 ± 0.002 | 0.938 ± 0.001 | 0.523 ± 0.002 | 0.942 ± 0.001 | 0.922 ± 0.001 | 0.500 ± 0.002 | 0.812 ± 0.002 | 0.812 ± 0.002 | 0.813 ± 0.002 |
| Human nonTATA Promoters | 0.919 ± 0.003 | 0.838 ± 0.004 | 0.867 ± 0.004 | 0.897 ± 0.003 | 0.890 ± 0.004 | 0.914 ± 0.003 | 0.780 ± 0.005 | 0.880 ± 0.004 | 0.880 ± 0.004 | 0.881 ± 0.004 |
| Human OCR Ensembl | 0.768 ± 0.003 | 0.710 ± 0.003 | 0.740 ± 0.003 | 0.760 ± 0.003 | 0.761 ± 0.002 | 0.786 ± 0.002 | 0.559 ± 0.003 | 0.749 ± 0.003 | 0.749 ± 0.003 | 0.750 ± 0.003 |

Table 8: Matthews Correlation Coefficient (MCC) scores for all individual tasks in the Genomics Benchmark.

| Model / Task | SSM-based | | Transformer-based | | | | | Hybrid | | |
|---|---|---|---|---|---|---|---|---|---|---|
| | HyenaDNA (436K) | Caduceus (470K) | DNABERT (86M) | DNABERT-2 (117M) | NT-v2 (500M) | Generator (1.2B) | Evo2 (7B) | HyenaMoE-F (116M) | HyenaMoE-S (131M) | HyenaMoE-E (145M) |
| Coding vs Intergenomic | 0.813 ± 0.004 | 0.810 ± 0.004 | 0.875 ± 0.003 | 0.846 ± 0.003 | 0.892 ± 0.003 | 0.915 ± 0.002 | 0.817 ± 0.004 | 0.930 ± 0.004 | 0.934 ± 0.004 | 0.939 ± 0.004 |
| Human vs Worm | 0.922 ± 0.003 | 0.922 ± 0.003 | 0.915 ± 0.003 | 0.930 ± 0.002 | 0.940 ± 0.002 | 0.950 ± 0.002 | 0.444 ± 0.005 | 0.951 ± 0.003 | 0.953 ± 0.003 | 0.955 ± 0.003 |
| Mouse Enhancers | 0.545 ± 0.055 | 0.000 ± 0.000 | 0.570 ± 0.051 | 0.143 ± 0.061 | 0.634 ± 0.048 | 0.646 ± 0.050 | 0.457 ± 0.053 | 0.648 ± 0.059 | 0.649 ± 0.059 | 0.651 ± 0.059 |
| Human Enhancers Cohn | 0.458 ± 0.011 | 0.472 ± 0.011 | 0.401 ± 0.011 | 0.492 ± 0.010 | 0.498 ± 0.010 | 0.496 ± 0.011 | 0.413 ± 0.011 | 0.495 ± 0.011 | 0.498 ± 0.011 | 0.500 ± 0.011 |
| Human Enhancers Ensembl | 0.676 ± 0.004 | 0.562 ± 0.005 | 0.694 ± 0.004 | 0.720 ± 0.004 | 0.714 ± 0.004 | 0.759 ± 0.004 | 0.352 ± 0.005 | 0.761 ± 0.004 | 0.762 ± 0.004 | 0.764 ± 0.004 |
| Human Regulatory | 0.805 ± 0.002 | 0.767 ± 0.002 | 0.904 ± 0.002 | 0.360 ± 0.003 | 0.910 ± 0.002 | 0.883 ± 0.002 | 0.310 ± 0.003 | 0.913 ± 0.002 | 0.915 ± 0.002 | 0.918 ± 0.002 |
| Human nonTATA Promoters | 0.850 ± 0.005 | 0.691 ± 0.008 | 0.754 ± 0.007 | 0.809 ± 0.006 | 0.794 ± 0.006 | 0.841 ± 0.006 | 0.580 ± 0.009 | 0.857 ± 0.007 | 0.859 ± 0.007 | 0.861 ± 0.007 |
| Human OCR Ensembl | 0.531 ± 0.005 | 0.438 ± 0.005 | 0.495 ± 0.005 | 0.542 ± 0.004 | 0.494 ± 0.004 | 0.560 ± 0.005 | 0.151 ± 0.005 | 0.565 ± 0.005 | 0.567 ± 0.005 | 0.569 ± 0.005 |

## C.4 ABLATION STUDY

HyenaLite significantly improves the efficiency of the original Hyena. As shown in Table 9, it reduces latency by more than $2\times$ and doubles the throughput while consuming slightly less GPU memory. These gains are achieved without sacrificing the core functionality of the Hyena operator, making HyenaLite a strong drop-in replacement whenever fast and lightweight sequence modeling is needed.

Table 9: Performance comparison between Hyena and HyenaLite modules.

| Metric | Hyena | HyenaLite | Improvement |
|---|---|---|---|
| Latency (ms) | 1.5 | 0.7 | $2.14\times$ faster |
| Throughput (tokens/s) | 1,328,840.43 | 2,758,400.37 | $2.08\times$ higher |
| Peak Memory (MB) | 80.44 | 72.42 | $\sim$10% less |

To further assess scalability and data efficiency, Figure 1 compares models by plotting average classification accuracy against pretraining data volume, with bubble size reflecting model parameter count. HyenaMoE variants (F, S, E) consistently attain state-of-the-art performance while using substantially less pretraining data and fewer parameters than large Transformer-based models such as NT and Gener*ator*. This highlights the superior data efficiency of our hybrid design, where the integration of Hyena filters and MoE routing enables strong generalization with significantly lower computational and data requirements.

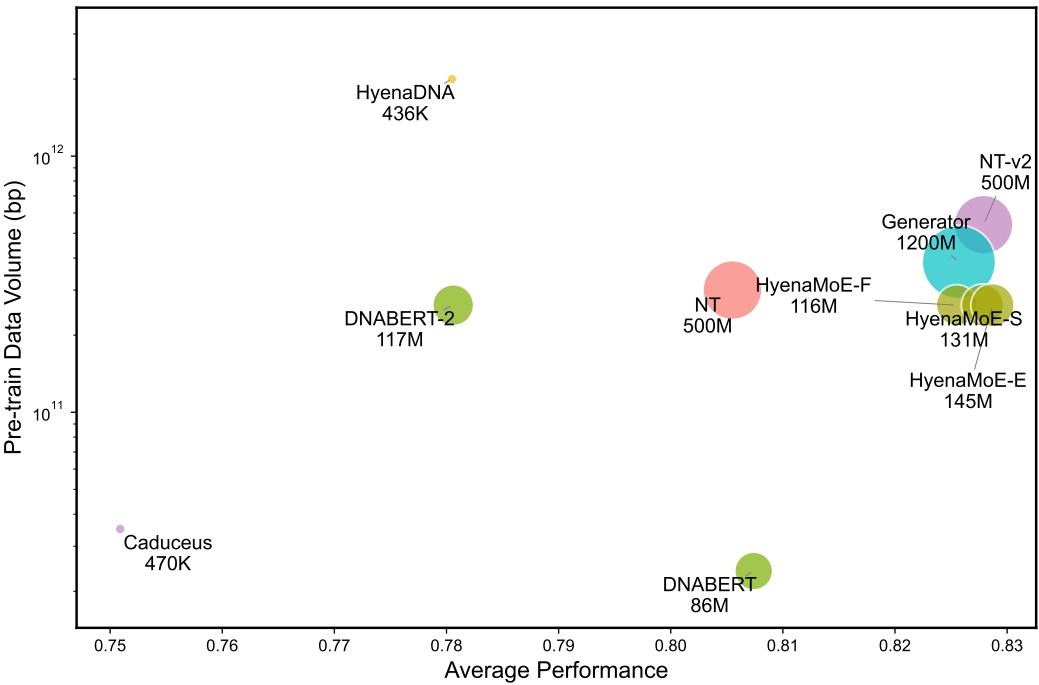

Figure 1: Relationship between model size, pretraining data volume, and average classification performance. HyenaMoE achieves strong accuracy and scalable generalization among all models.

Figure 2 shows $R^2$ scores as a function of input sequence length. While most models exhibit gradual improvement as the context window expands, HyenaMoE consistently achieves superior performance across all tested lengths, highlighting its robust generalization capability over a wide range of input scales. Notably, even at shorter lengths (e.g., 512 or 1,024), HyenaMoE already leads, and it continues to improve with longer inputs. In contrast, DNABERT is limited to an input length of 2,048 due to its architectural design. These results further underscore the scalability and adaptability of HyenaMoE for long-context genomic modeling, outperforming both structured state-space and conventional Transformer-based baselines.

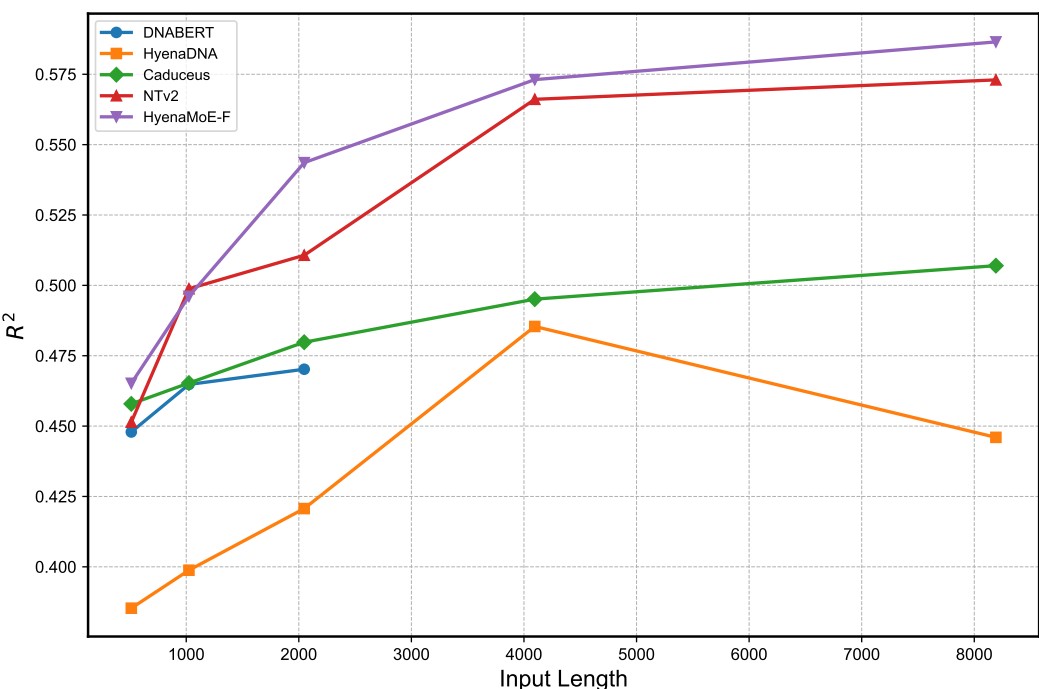

Figure 2: $R^2$ performance across input lengths from 512 to 8192. HyenaMoE achieves strong accuracy and scalable generalization among all models.

## D LLM Usage Statement

In preparing this work, large language models (LLMs) were used only as auxiliary tools to improve the presentation of the manuscript. They assisted with grammar correction, language refinement, and occasional suggestions for clearer phrasing and structure, but did not generate or alter the technical content of the paper. All research ideas, methods, experiments, analyses, and conclusions were fully conceived, implemented, and validated by the authors. The authors have carefully verified all LLM-assisted text to ensure accuracy, originality, and compliance with the ICLR Code of Ethics. LLMs were not considered contributors or co-authors of this work.