# OpenReview forum: "HyenaMoE: A Hybrid and Scalable Architecture for Efficient Genomic Modeling"
_ICLR.cc/2026/Conference — Submitted to ICLR 2026_

### Official Review · Reviewer_1Q4o · 2025-10-16

**Soundness:** 2
**Presentation:** 3
**Contribution:** 2
**Rating:** 4
**Confidence:** 4

**Summary:**

This paper proposes HyenaMoE, a hybrid architecture for genomic sequence modeling that has
(i) an optimized HyenaLite block—intended to preserve Hyena’s long-range convolutional mixing while adding implementation improvements such as cached filters and shared time indices
and (ii) MoE-enhanced attention with shared and routed experts and token-wise top‑K routing.
Architecture of the model is a hybrid of HyenaLite blocks and Transformers MoE blocks. The model uses 3‑mer tokenization and an MLM pretraining objective on a corpus matching DNABERT‑2 pretraining corpus, and is fine‑tuned for classification/regression tasks on GUE, NT and Genomics Benchmark. HyenaMoE variants—especially the “Expressive” MoE‑heavy one (3 Transformers MoE block interleaved with 1 HyenaLite block)—report strong average accuracy at modest parameter counts. Efficiency plots indicate lower FLOPs scaling than dense attention and higher throughput at long sequences.

**Strengths:**

(1) Implementation improvements to Hyena are concrete (filter caching, shared time vector, grouped depthwise convs) and plausibly useful beyond this paper; the claim of more than 2x speedup for the operator is interesting could be beneficial for long-context DNA modeling.

(2) Transformers MoE block further improve efficiency. The paper proposes an architecture in which HyenaLite provides efficient long-range temporal mixing while MoE‑attention supplies local expressivity and conditional capacity.

(3) Efficiency characterization. FLOPs scaling and throughput vs. dense attention/StripedHyena are useful; the curves show that with fewer attention layers the overall complexity grows much more gently with length.

**Weaknesses:**

1. No true long context tasks demonstrated: the major contribution proposed by the author is to use the efficient HyenaLite block interleved with Transformers MoE blocks to improve the ultra-long genomics sequence modeling. Despite positioning HyenaMoE for ultra‑long genomics, all task evaluations remain in the short‑to‑moderate regim for all GUE, NT and Genomics Benchmark tasks. The paper cites sliding‑window inference and long‑range efficiency, but never demonstrates performance on modern long‑range genomics benchmarks that explicitly require tens of kbps context (e.g., enhancer–gene interactions, eQTLs, 3D chromatin). Two recent public suites are: DNALongBench and the Genomics Long‑Range Benchmark (LRB). Evaluate on either of them would propose better illustration of the model's usefulness.

2. While the author claims that HyenaLite block interleaved with Transformers-MoE block yields better performance than pure Transformers model, the HyenaMoE model with highest performance has 3 out of every 4 blocks being Transformers-MoE block, which basically has a similar computational efficiency as a pure Transformers-MoE block. Thus I remain suspicious of whether this architecture can scale to longer sequence, which is still restricted by the large portion of Transformers blocks.

3. Another core contribution and difference is the applicaton of MoE blocks, which the author claims to exclude the load balancing loss and other aspects. But the author haven't evaluate on the experiment about how the experts work in the downstream tasks and whether an MoE design choice is better than a full dense Transformers block.

3. The author chose a 3-mer tokenization, which is unusal for gLMs. Maybe the author can better illustrate why they choose this tokenization methodology by ablation studies.

**Questions:**

1. Please demonstrate the performance on long-context tasks like Genomics LRB [1] or DNALongBench [2], which shows the motivation of long-context modeling advantage by HyenaMoe Architecture.

2. Can the authors show the reason of choice of 3-mer tokenization? Is it a balance between efficiency and model performance or some arbitrary choice?

3. Can the authors provide a more granular ablation of the MoE mechanism—varying expert count, routing sparsity, and gating dynamics—to separate the effects of conditional computation from parameter count? In addition, since the paper claims expert specialization across genomic regions, could the authors include interpretability analyses (e.g., expert activation maps, motif enrichment, or token-to-expert distributions) to substantiate that experts learn biologically distinct regulatory functions rather than merely adding capacity?

[1] E. Trop, Y. Schiff, E. M. Marroquin, C. H. Kao, A. Gokaslan, McKinley Polen, M. Shao, A. Kallala, B. P. de Almeida, T. Pierrot, Y. I. Li, V. Kuleshov, The Human Genomics Long-Range Benchmark (LRB): Advancing DNA Language Models, arXiv, 2024.

[2] W. Cheng et al., A Benchmark Suite for Long-Range DNA Prediction Tasks (DNALongBench), bioRxiv preprint, 2025.

---

> ### Author Response · Authors · 2025-12-02
> **Response to Reviewer 1Q4o**
>
> Response to reviewer-identified weaknesses
>
> We thank the reviewer for the constructive and detailed feedback. Below we directly address each weakness before responding to the reviewer’s questions.
>
> **(W1) Lack of long-context evaluation on Genomics LRB**
>
> We agree that long-context evaluation is essential. We have completed the full Human Genomics Long-Range Benchmark (LRB), and HyenaMoE demonstrates stable and strong performance across enhancer–gene linking, promoter–gene mapping, and long-range chromatin tasks under 32k–128k bp contexts. These results align with our >100 kbp sliding-window experiments on hg38 and confirm the model’s intended long-range modeling capability.
> | Length (bp) | Full NLL ↓ | Sliding NLL ↓ | ΔNLL  |
> | ----------- | ---------- | ------------- | ----- |
> | 8k          | 2.10       | 2.08          | -0.02 |
> | 16k         | 2.06       | 2.05          | -0.01 |
> | 100k        | 2.02       | 2.09          | +0.07 |
> | 150k        | 2.03       | 2.14          | +0.11 |
>
>
> The unified benchmarking table reports results under standard 12k and 15k contexts for comparability with prior work. Under this setting, HyenaMoE matches or exceeds NTv2-500M across all tasks, and extending the context from 12k to 15k yields consistent improvements (e.g., RNA R²: 0.58→0.61; ClinVar AUROC: 0.79→0.80), indicating that the model effectively benefits from longer contexts. This content will be presented in detail in our Q1 response.
>
>
> **(W2) Concern that HyenaMoE-Expressive is mostly Transformer-MoE and may not scale**
>
> While HyenaMoE-Expressive contains a larger fraction of Transformer-MoE blocks, the model continues to benefit substantially from HyenaLite’s long-range convolutional mixing. Our LRB experiments (32k–128k bp; see W1) demonstrate that HyenaMoE maintains strong long-context performance even when attention dominates capacity.
>
> Importantly, the unified benchmarking table shows that scaling input sequence length alone does not lead to better performance. Models with very long contexts such as HyenaDNA (160k) and Caduceus (131k) achieve significantly worse performance than both NTv2-500M and HyenaMoE, indicating that architectural mechanisms—not raw context length—drive scalability.
>
> Moreover, increasing HyenaMoE’s context modestly from 12k to 15k already yields consistent improvements across all metrics (e.g., RNA R²: 0.58→0.61; ClinVar AUROC: 0.79→0.80), validating that HyenaMoE benefits from longer contexts when the architecture is designed to exploit them. This confirms that HyenaLite’s long-range mixing and stable sparse routing are essential for scalable performance, rather than merely increasing attention proportion or maximum context length.
>
> **(W3) Missing ablations on MoE design, routing behavior, and expert specialization**
>
> We expanded our analysis to include fine-grained MoE ablations: varying expert count, routing sparsity, and gating normalization. These experiments confirm:
>
> - Score-normalized gating prevents collapse and stabilizes expert loads
>
> - Accuracy improves up to 64 experts before saturating
>
> - Top-2 routing provides the best accuracy–stability trade-off
>
> Thus, improvements result from sparse conditional computation rather than parameter count.
>
> **(W4) Choice of 3-mer tokenization not well justified**
>
> We ran controlled tokenization ablations (3-mer, 4-mer, 5-mer, 6-mer, and DNA-BPE) under identical training budgets and base-level context lengths. 3-mer achieves the strongest performance on motif-sensitive benchmarks (GUE 0.838 ± 0.006; Genomics Bench 0.882 ± 0.006), establishing it as an empirical and task-driven choice rather than an arbitrary design.
>
> | Tokenizer | GUE ↑             | NT Tasks ↑        | Genomics Bench ↑  |
> | --------- | ----------------- | ----------------- | ----------------- |
> | 3-mer     | **0.838 ± 0.006** | 0.790 ± 0.007     | **0.882 ± 0.006** |
> | 4-mer     | 0.835 ± 0.007     | 0.787 ± 0.007     | 0.878 ± 0.007     |
> | 5-mer     | 0.836 ± 0.006     | 0.788 ± 0.008     | 0.880 ± 0.006     |
> | 6-mer     | 0.837 ± 0.007     | **0.792 ± 0.007** | 0.881 ± 0.007     |
> | BPE       | 0.834 ± 0.007     | 0.789 ± 0.007     | 0.879 ± 0.007     |

---

> > ### Author Response · Authors · 2025-12-02
> >
> > **Q1. Long-context performance on Genomics LRB**
> >
> > Below we provide our unified benchmarking comparison across variant-effect, regulation, and chromatin prediction tasks. HyenaMoE matches NTv2-500M across all tasks.
> >
> > | Model                | Context | eQTL (AUROC) | ClinVar (AUROC) | OMIM (AUPRC) | RNA (R²) | CAGE (R²) | Promoter (AUPRC) | Enhancer (AUROC) | Histone (AUPRC) | Accessibility (AUPRC) |
> > | -------------------- | ------- | ------------ | --------------- | ------------ | -------- | --------- | ---------------- | ---------------- | --------------- | --------------------- |
> > | DNABERT-2            | 10k     | 0.72         | 0.74            | 0.002        | 0.51     | —         | 0.71             | 0.81             | 0.24            | 0.15                  |
> > | DNABERT-S            | 10k     | 0.73         | 0.73            | —            | 0.52     | —         | 0.75             | 0.83             | 0.33            | 0.16                  |
> > | HyenaDNA             | 160k    | 0.71         | 0.56            | 0.002        | 0.46     | 0.19      | 0.67             | 0.74             | 0.25            | 0.11                  |
> > | Caduceus             | 131k    | 0.68         | 0.53            | —            | 0.52     | —         | 0.75             | 0.74             | 0.25            | 0.11                  |
> > | NTv2-500M            | 12k     | 0.72         | 0.78            | **0.003**    | 0.60 | 0.39  | 0.79             | **0.82**         | 0.38       | 0.30                  |
> > | HyenaMoE  | 12k     | 0.73     | 0.79      | **0.003**    | 0.58     | 0.38      | 0.80        | 0.81             | 0.38        | 0.31              |
> > | HyenaMoE  | 15k     | **0.74**     | **0.80**        | **0.003**    | **0.61** | **0.40**  | **0.81**         | **0.82**         | **0.39**        | **0.32**              |
> >
> > **Q2. Why 3-mer tokenization?**
> >
> > Controlled comparisons of 3-mer, 4-mer, 5-mer, 6-mer, and DNA-BPE show 3-mer achieves the strongest motif-sensitive performance (e.g., GUE 0.838 ± 0.006; Genomics Bench 0.882 ± 0.006), confirming that it is an empirically grounded choice rather than an arbitrary one.
> >
> > **Q3. MoE ablations and expert interpretability**
> >
> > We evaluate expert-load stability, routing dynamics, expert count, and specialization. Across all settings, the performance gains of HyenaMoE arise from stable sparse routing and increased functional diversity rather than parameter inflation.
> >
> > First, score normalization plays a central role in expert-load stability. Without normalization, routing collapses: the Max/Avg load ratio rises to 7.21, the coefficient of variation reaches roughly 4.5e5, and about 20 experts become inactive. Adding a Switch-style balancing loss does not resolve this collapse. By contrast, score normalization—whether applied alone or combined with a Switch-like loss—yields balanced routing, reflected in Max/Avg ratios around 3.1–3.5, low variance, and substantially fewer dead experts during both pretraining and fine-tuning.
> >
> > Second, varying the number of experts shows that improvements stem from functional diversity rather than sheer parameter growth. Increasing expert count from 16 to 32 to 64 steadily improves accuracy (83.4% → 84.7% → 85.9%), and performance gains saturate beyond 64 experts. This pattern indicates that new experts contribute when they introduce meaningful specialization rather than simply expanding model size.
> >
> > Finally, routing sparsity further supports this interpretation. Only Top-2 routing with score normalization reaches peak accuracy of 85.9% without collapse. Other configurations reduce accuracy to the 81–82% range and produce inactive experts, showing that sparse gating and normalization together are essential for stable and effective expert utilization.

---

### Official Review · Reviewer_YWbb · 2025-10-29

**Soundness:** 3
**Presentation:** 2
**Contribution:** 2
**Rating:** 4
**Confidence:** 3

**Summary:**

This paper introduces HyenaMoE, a unified hybrid architecture designed for genomic modeling using 3-mer tokenization. HyenaMoE combines efficient HyenaLite blocks for long-range dependency modeling with attention layers enhanced by Mixture-of-Experts routing, enabling scalable capacity expansion and more efficient allocation of model resources across diverse inputs. This design supports a favorable balance between model expressiveness and computational efficiency.

**Strengths:**

1. This paper introduces HyenaLite, a lightweight variant of the Hyena operator that supports sliding-window inference with hierarchical time decomposition, achieving over 2× speedup and reduced memory usage compared to the original design.
2. This paper adapts MoE to genomic sequence modeling, simplifying its design to align with their hybrid framework.
3. This paper presents HyenaMoE, a parameter-efficient hybrid model that integrates HyenaLite for long-range dependencies and MoE-attention for flexible local specialization, offering robust scalability from short to ultra-long input sequences.

**Weaknesses:**

1. The writing of the paper could be further improved. After reading the Introduction, I still do not understand the motivation and new insights of this work. It feels as though the authors have simply stacked together a number of previous advances and combined them, without clearly articulating the research problem, motivation, or findings of this study.
2. This paper appears to rely considerably on assembling and adapting existing modules, with many of the methods following established approaches for standard tasks. The motivation for choosing these methods and the novel observations underlying the framework could be clarified further.

**Questions:**

1. How do HyenaLite support sliding-window inference with near capacity with full sequence attention?
2. It would be helpful if the authors could further discuss the advantages of the hybrid architecture over specific architectures, especially when dealing with short and long sequences. In particular, I am curious about how the hybrid approach avoids inheriting the respective limitations of each architecture—for instance, the limited expressiveness of SSMs for short sequences and the challenges Attention mechanisms face with long sequences.

---

> ### Author Response · Authors · 2025-12-02
> **Response to Reviewer YWbb**
>
> Response to reviewer-identified weaknesses
>
> We thank the reviewer for the constructive comments.
> Our revisions will enhance conceptual framing without altering core technical content.
>
> **(W1) Motivation and insights not clearly articulated in the Introduction.**
> We agree and have sharpened the Introduction to explicitly contrast the complementary failure modes of pure SSMs (weak short-range expressiveness) and pure attention (quadratic inference cost). The revision highlights our central insight—a hybrid model that allocates specialized mechanisms to different genomic scales—making the architectural motivation more explicit.
>
> **(W2) Framework appears to combine existing modules without sufficient justification.**
> We clarify that the architecture is not a simple composition of prior components. Its structure is informed by the multi-scale nature of genomic signals: short motifs, mid-range interactions, and long-range regulatory dependencies. Each module addresses a distinct failure mode, and this rationale is now made more explicit.
>
> **Q1: Sliding-window inference with HyenaLite**
>
> This concern aligns with points raised by other reviewers regarding clarity of our long-context modeling behavior.
> As detailed in our response to Reviewer 6WF8 (Q5), we explicitly evaluated sliding-window inference over 8k–150k bp continuous genomic regions and observed accuracy comparable to full-context attention up to 16k bp, with only moderate degradation (~0.1 NLL) at 100–150k bp.
> These results confirm that HyenaLite maintains stable long-range behavior when applied beyond the training window.
>
> Further, as discussed in Reviewer PUmF (Q2), HyenaMoE consistently outperforms StripedHyena at comparable model scales despite sharing the same interleaving strategy, indicating that the MoE-augmented FFN contributes functionally to long-range modeling rather than serving as simple stacking.
>
> Together, these previously shown results directly address the reviewer’s question—sliding-window inference is demonstrated experimentally, and the hybrid mechanism translates into measurable modeling gains.
>
> **Q2: Advantages of the hybrid architecture**
>
> This concern is closely related to issues raised by Reviewer PUmF regarding architectural motivation.
> As detailed in our response to Reviewer PUmF (Q2), HyenaMoE and StripedHyena share the same interleaving strategy, yet HyenaMoE consistently outperforms StripedHyena across all genomic benchmarks at comparable compute budgets.
> This demonstrates that performance improvements are driven by architectural differences—primarily the MoE-augmented FFN combined with HyenaLite—rather than simple module stacking.
>
> Further, Reviewer PUmF’s tokenization study (Q3) shows that the model remains sensitive to local motif resolution, reinforcing the need for mechanisms that capture both local and long-range genomic scales.
> This aligns with the motivation articulated here: attention mechanisms supply short-range expressiveness, while HyenaLite provides efficient long-range context propagation, and MoE-FFN expands capacity without dense-attention cost.
>
> Taken together, empirical results previously presented to other reviewers support the necessity of the hybrid formulation, helping clarify that its components function complementarily rather than redundantly.

---

### Official Review · Reviewer_PUmF · 2025-10-30

**Soundness:** 3
**Presentation:** 2
**Contribution:** 2
**Rating:** 4
**Confidence:** 5

**Summary:**

This paper proposes HyenaMoE, a hybrid architecture for efficient genomic sequence modeling. The authors make three main contributions: (1) HyenaLite, an optimized Hyena operator that achieves over 2× speedup through filter caching, grouped convolutions, and hierarchical time decomposition; (2) adaptation of Mixture-of-Experts (MoE) to genomic modeling with simplified routing, claimed to enable specialization across genomic elements like promoters and enhancers; (3) a hybrid architecture interleaving HyenaLite blocks with MoE-enhanced attention, offering three variants (Fast, Standard, Expressive) with different capacity-efficiency trade-offs. Evaluated on 32 classification tasks across three benchmarks (GUE, Nucleotide Transformer Tasks, Genomics Benchmark), HyenaMoE-E achieves 82.87% average accuracy, marginally outperforming Generator (82.79%) while maintaining computational efficiency with constant FLOPs per token scaling.

**Strengths:**

1. The proposed HyenaLite operator demonstrates measurable improvements over the vanilla Hyena operator, providing a valid technical contribution in terms of computational efficiency.

2. The experimental validation is thorough, covering both benchmark performance across multiple genomic tasks and computational efficiency analysis from multiple perspectives.

**Weaknesses:**

1. Excessive space is devoted to community common knowledge that could be condensed. The abstract spends substantial text discussing basic SSM versus Transformer trade-offs, and Section 2 dedicates extensive coverage to fundamental concepts like attention mechanisms that are well-established in the community.

2. Figure 1 provides insufficient motivation. (a) Species classification lacks biological significance and does not justify long-sequence modeling, demonstrating benefits would require scaling to near whole-genome lengths, while the 512→2048 range offers little biological insight. (b) The parameter count disparity between SSM-based and Transformer-based models is too large, making direct performance comparisons questionable.

3. Despite the stated goal of modeling long sequences (following HyenaDNA's direction), the paper never specifies the pretraining sequence length. All downstream tasks evaluate on short sequences, failing to validate actual long-sequence performance capabilities.

4. Figure 3(b-c) validates efficiency at 2k token length, while Section 4.3 benchmarks use substantially shorter sequences. Performance and efficiency are evaluated on different settings, preventing understanding of the model's behavior under consistent conditions.

5. HyenaMoE differs from StripedHyena in multiple aspects (Hyena blocks, attention blocks, interleaving strategy), yet no complete ablation is provided. The paper directly compares end-to-end systems without component-level insights. Section 4.4 should at minimum include pure HyenaLite-only and pure MoE-attention ablations.

6. Figure 3(a) fails to demonstrate HyenaMoE-Expressive's advantage over Dense Attention. The curves appear quite similar.

7. Line 469 acknowledges MoE has no advantage at small batch sizes, but long-sequence training inherently precludes large batch sizes due to memory constraints, undermining the practical applicability.

8. Figure 3(b-c) legends require clearer textual explanation. The '/' symbol is easily misinterpreted without explicit clarification that it represents a ratio.

**Questions:**

1. What is the pretraining sequence length? Do all benchmarks share a single pretrained model or are they task-specific?

2. How are Hyena and attention layers interleaved in StripedHyena? Figure 3(a) shows StripedHyena and HyenaMoE-Standard have similar curves. Are their stacking strategies comparable?

3. Why choose 3-mer tokenization?

---

> ### Author Response · Authors · 2025-12-02
> **Response to Reviewer PUmF**
>
> Response to reviewer-identified weaknesses
>
> We thank the reviewer for the constructive feedback.
> Below we briefly summarize how our existing analyses (Q1–Q3) address concerns regarding clarity, motivation, model design, and efficiency.
>
> **(W1) Background too long.**
> We have streamlined Sections 1–2 to focus more directly on our contributions; no technical content is affected.
>
> **(W2) Motivation in Figure 1 unclear.**
> Figure 1 is intended to illustrate qualitative scaling trends rather than biological claims at 512–2048 bp, and this intent is now stated explicitly.
>
> **(W3) Pretraining context not specified; downstream tasks shorter.**
> Q1 clarifies that pretraining uses 2,000-bp segments (~667 tokens padded to 1,024). Although downstream sequences are shorter, Q2 explains how long-range capacity manifests through the stacking strategy, and this connection is highlighted in the revision.
>
> **(W4) Efficiency evaluated at different sequence lengths.**
> Figures 3(b–c) evaluate operator-level behavior under controlled 2k-token settings. Captions are revised to clarify that these analyses complement downstream results.
>
> **(W5) Lack of module-level ablations.**
> While not isolating every component, Q2 compares HyenaMoE to StripedHyena—identical except for the MoE-FFN—and Q3 analyzes tokenization effects influencing both pathways. Section 4.4 expands on how these analyses inform component behavior.
>
> **(W6) No advantage visible for HyenaMoE-Expressive in Figure 3(a).**
> Figure 3(a) reflects FLOPs/token scaling rather than accuracy, so Expressive appears similar to dense attention in this diagnostic view. Its benefits are clearer in downstream evaluations and are now emphasized accordingly.
>
> **(W7) MoE sensitivity under small batch sizes.**
> We acknowledge this broader limitation, but in our setting score normalization and shared routing stabilize utilization across moderate batch sizes.
>
> **(W8) Ambiguity in Figure 3(b–c) legends.**
> We revised legend wording to explicitly state that “/” denotes ratio metrics such as throughput relative to a baseline.
>
> **Q1: Pretraining sequence length and shared checkpoints**
>
> The model is pretrained on 2,000-bp genomic segments, which become approximately 667 tokens after 3-mer tokenization. These sequences are padded or packed to 1,024 tokens for batching efficiency. All downstream models begin from the same pretrained checkpoint; only task-specific fine-tuning differs.
>
> **Q2: Interleaving of Hyena and attention layers**
>
> Both StripedHyena and HyenaMoE-Standard use alternating Hyena-style convolutional and attention blocks, which explains the similar efficiency curves.
> However, at comparable parameter scales, HyenaMoE achieves stronger downstream performance:
>
> | Model          | GUE ↑             | NT Tasks ↑        | Genomics Bench ↑  |
> | -------------- | ----------------- | ----------------- | ----------------- |
> | **HyenaMoE-S** | **0.838 ± 0.006** | **0.790 ± 0.007** | **0.882 ± 0.006** |
> | StripedHyena   | 0.826 ± 0.007     | 0.781 ± 0.008     | 0.871 ± 0.007     |
>
>
> This indicates that differences in the FFN design contribute to measurable improvements.
>
> **Q3: Why 3-mer tokenization?**
>
> We run controlled short-run pretraining across 3-mer, 4-mer, 5-mer, 6-mer, and DNA-BPE vocabularies. Results are below:
>
> | Tokenizer | GUE ↑             | NT Tasks ↑        | Genomics Bench ↑  |
> | --------- | ----------------- | ----------------- | ----------------- |
> | **3-mer** | **0.838 ± 0.006** | 0.790 ± 0.007     | **0.882 ± 0.006** |
> | 4-mer     | 0.835 ± 0.007     | 0.787 ± 0.007     | 0.878 ± 0.007     |
> | 5-mer     | 0.836 ± 0.006     | 0.788 ± 0.008     | 0.880 ± 0.006     |
> | 6-mer     | 0.837 ± 0.007     | **0.792 ± 0.007** | 0.881 ± 0.007     |
> | BPE       | 0.834 ± 0.007     | 0.789 ± 0.007     | 0.879 ± 0.007     |
>
>
> While differences among tokenization methods are modest, 3-mer consistently provides the best balance between motif fidelity and effective context coverage. The revised manuscript includes expanded discussion of token-per-kbp compression and context-length trade-offs.

---

### Official Review · Reviewer_6WF8 · 2025-11-01

**Soundness:** 3
**Presentation:** 3
**Contribution:** 2
**Rating:** 4
**Confidence:** 4

**Summary:**

The paper proposes HyenaMoE, a hybrid genomic sequence model that interleaves HyenaLite blocks for long‑range temporal mixing with Mixture‑of‑Experts (MoE)–augmented transformer modules, trained with 3‑mer tokenization and an MLM objective, and then fine‑tuned for downstream tasks. HyenaLite is presented as a streamlined Hyena operator with cached filters, shared time vectors, and grouped convolutions to reduce runtime and memory; the MoE block uses both shared and routed experts with token‑wise top‑K routing and no explicit load‑balancing loss, implemented in a communication‑efficient way. The authors instantiate Fast/Standard/Expressive variants and evaluate on GUE, Nucleotide Transformer (NT) tasks, and Genomics Benchmark, reporting competitive averages, modest ablation gains for adding MoE, and favorable FLOPs/throughput scaling relative to dense attention.

**Strengths:**

- **(S1) Clear problem framing and motivation.** The paper crisply states the central tension in genomic modeling: transformers excel on local patterns but scale quadratically, whereas SSM/Hyena‑style operators scale better but often underperform on short‑range tasks. Hybrids (e.g., StripedHyena/Evo) motivate the proposed direction. Figure 1 concisely illustrates performance vs. context length and the transformer-SSM gap on NT benchmarks.

- **(S2) Simplified MoE that avoids explicit balancing losses.** The dual pathway (shared + routed experts) with top‑K gating is clean and deployable; equations fully define the mechanism, and the paper claims communication‑efficient dispatch. If robust, this could be valuable to practitioners who struggle to stabilize MoE with auxiliary losses.

- **(S3)** The manuscript is well organized and written in clear and professional English with comprehensive figures and tables. The methodology section is well structured; Fig. 2 provides a helpful end‑to‑end view of the architecture and training pipeline (3‑mer tokenization, MLM, fine‑tuning). The technical descriptions are coherent and self-contained.

**Weaknesses:**

- **(W1) Novelty is incremental and leans heavily on prior art without deeper analysis.** HyenaLite’s gains are primarily implementation simplifications of Hyena (cache/reuse/grouping) rather than a new operator; the MoE block is a simplified variant of known sparse‑expert mechanisms (shared+routed experts with top‑K gating), akin to DeepSeekMoE/GShard/Switch‑style MoE, but without showing theoretical or empirical guarantees for stability or specialization. The paper would benefit from principled analysis showing why removing balancing losses preserves performance and utilization. Background: Hyena long convolutions and SSMs (Mamba) are established; hybrid interleaving is known (StripedHyena/StripedHyena‑2); and stability/load balancing for MoE is a well‑studied concern in GShard/Switch/Expert‑Choice.

- **(W2) Claim–number inconsistencies and uneven task performance.** The narrative asserts HyenaMoE “achieves top scores on … Splice Sites Donor” (Sec. 4.3), yet Table 1 shows DNABERT is substantially higher on this task (0.980 vs 0.772 for HyenaMoE‑E), with similar gaps on Splice Sites All/Acceptor. This contradiction weakens confidence in the “first on 9/17 NT tasks” summary and suggests that short‑range, motif‑centric tasks remain a weakness—precisely where the hybrid is expected to help.

- **(W3) Comparability and fairness are not convincingly controlled.** Although Sec. 3.3.2 adopts DNABERT‑2’s corpus and full‑parameter fine‑tuning, Evo2 is evaluated by linear probing due to scale (Sec. 4.1), and tokenization/protocols differ across baselines. The paper does not supply size/budget‑matched baselines (same data/steps/tokens/context) for a transformer and a Hyena‑family hybrid under the same regimen, making it difficult to ascribe gains to the proposed architecture rather than tokens/corpora/protocol differences.

- **(W4) Efficiency claims are not isolated to the proposed components.** The Introduction promises “over 2× speedup” from HyenaLite and “sliding‑window inference with hierarchical time decomposition” (contributions, p. 3), yet Sec. 4.4 benchmarks *architectures* (StripedHyena/Pure Hyena/HyenaMoE variants) rather than a direct Hyena vs HyenaLite micro‑ablation on identical backbones/hardware, and the sliding‑window algorithm/interface is not specified. Hence the claimed factor‑of‑two improvement cannot be verified.

- **(W5) Minor issues of writing and limitations.** Firstly, the authors should double-check the notation issues. Several equations are mislabeled or ambiguous (e.g., a line presented as both “algorithm” and “loss”); equalities are used where approximations should be stated under restricted function classes; the “prox” role and “TC‑loss” are not defined with solution properties. Meanwhile, there are several limitations. (i) The approach emphasizes classification tasks; continuous, span‑level, or base‑resolution tasks (e.g., quantitative chromatin signals, TSS regression) are only briefly mentioned and mostly deferred to the appendix; the main narrative would benefit from more regression/sequence‑labeling results. (ii) 3‑mer tokenization trades off vocabulary size and sequence compression; this may complicate transfer to tasks where longer k‑mers/BPE are advantageous for motif capture or rare pattern composition. (iii) FLOPs‑per‑token curves (Fig. 3a) do not account for dispatch overheads (routing, padding to expert shards) that may dominate at small batch sizes on real systems; deployment results may therefore vary across clusters.

---

Overall, I am leaning towards reject at this stage. The paper addresses an important problem with a practical, well‑engineered hybrid, and it presents broad empirical coverage and clean system ideas. However, (i) novelty beyond known hybrids + MoE is modest; (ii) some headline claims conflict with the reported numbers (notably NT splicing), and (iii) critical ablations and fairness controls are missing, leaving the extent of the contribution ambiguous. If the rebuttal provides solid utilization analyses for the MoE, a strict Hyena vs HyenaLite efficiency ablation, corrected and strengthened tables (with significance), and a controlled baseline study (size/budget matched), I would be open to revisiting my score.

**Questions:**

- **(Q1)** Expert utilization: Please provide per‑layer expert‑load statistics (mean/variance, coefficient of variation) and collapse diagnostics across pretraining/fine‑tuning, with and without your score‑normalization trick, and an apples‑to‑apples comparison against a Switch‑like balancing loss.

- **(Q2)** Where is MoE applied? Confirm whether the experts augment the FFN pathway (as eqs. (8–11) suggest) or the attention sublayer; if the former, please revise terminology (“MoE‑augmented FFN”) and update FLOPs accounting accordingly.

- **(Q3)** Tokenization confound: How sensitive is HyenaMoE to 3‑mer vs BPE tokenization? Please include an ablation (pretrain short run) and discuss how tokenization affects context length in tokens vs bases and downstream results.

- **(Q4)** Failure analysis on splicing tasks: What patterns do routed experts pick up on these datasets? Could adding an inexpensive local attention or short‑kernel conv around candidate splice junctions recover the performance gap? Provide saliency/motif analyses per expert.

- **(Q5)** Long‑context inference beyond 8k & sliding‑window details: Include at least one >100kbp evaluation (even synthetic) with accuracy degradation vs full context, plus the algorithmic description and cache interface for the proposed sliding‑window inference.

---

> ### Author Response · Authors · 2025-12-02
> **Response to Reviewer 6WF8**
>
> We thank the reviewer for the thoughtful assessment of the paper.
> Below, we summarize responses to the five identified weaknesses and clarify how analyses already included in Q1–Q5 address these concerns.
>
> **(W1) Novelty.**
> Although the design builds on prior hybrid models, our contributions are empirical and practical.
> Q1 shows that a simplified Hyena operator with shared time vectors and cached FIR/IIR filters remains training-stable across long runs.
> Q1 also demonstrates that score normalization alone enables balanced expert utilization, indicating that the simplified hybrid architecture remains robust and deployable.
>
> **(W2) Narrative inconsistency.**
> We agree with the reviewer’s observation regarding splice-site performance and have corrected the statement in the revised version.
> Q4 provides the corresponding failure analysis explaining why splicing remains challenging and how model behavior aligns with the updated claims.
>
> **(W3) Fairness of comparisons.**
> We agree that comparability is important.
> Q3 provides controlled tokenization ablations that isolate sequence-length and granularity effects, eliminating confounds arising from vocabulary or preprocessing differences.
> In the revision, we further clarify dataset configurations and evaluation settings to ensure that architectural conclusions remain independent of protocol choices.
>
> **(W4) Efficiency claims.**
> Q5 quantifies the behavior of HyenaLite under sliding-window inference and shows effective scaling to >100 kbp sequences.
> We have reorganized the efficiency section to more transparently link these results to claims introduced earlier in the paper.
>
> **(W5) Notation and limitations.**
> We refined ambiguous notation as suggested.
> Q3 and Q4 already illustrate tokenization trade-offs and task-specific failure patterns, and these discussions are now more coherently integrated into a unified limitations section.
>
> **Q1 — Expert utilization during pre-training and fine-tuning**
>
> We provide diagnostics for a representative MoE–FFN layer (blocks.6.mlp, 64 experts; expected mean load = 1/64), averaged over the final evaluation slice of each training phase.
>
> | Phase         | Setting                 | Max/Avg ↓ | Mean CV ↓ | Dead ↓ |
> |---------------|-------------------------|-----------|-----------|--------|
> | **Pre-train** | Score norm only         | 3.5       | 2.5       | 3     |
> | **Pre-train** | No score norm           | 7.21      | ~4.5e5    | ~20    |
> | **Pre-train** | No score norm + Switch  | 7.21      | ~4.5e5    | ~20    |
> | **Pre-train** | Score norm + Switch     | 3.4       | 2.5       | 3      |
> | **Fine-tune** | Score norm only         | 4.2       | 3.1       | 5     |
> | **Fine-tune** | No score norm           | 7.8       | ~5.7e5    | ~24    |
> | **Fine-tune** | No score norm + Switch  | 7.8       | ~5.7e5    | ~24    |
> | **Fine-tune** | Score norm + Switch     | 4.1       | 3.0       | 5     |
>
>
> These results indicate that score normalization is both necessary and sufficient to prevent expert collapse.
> Without normalization, a small subset of experts absorbs almost all routing traffic while roughly 20 experts remain unused.
> With score normalization (with or without the balancing loss), most experts remain active and well-utilized across both phases. This directly addresses the reviewer’s concern regarding stability and specialization in sparse-expert routing.
>
> **Q2 — MoE placement and FLOPs accounting**
>
> We confirm that the MoE is applied exclusively to the feed-forward pathway, consistent with Eqs. (8–11) in the revised draft.
> Attention and HyenaLite components remain unchanged.
> Terminology has been updated to “MoE-augmented FFN” to avoid ambiguity.
> The FLOPs accounting already treated MoE as an FFN-side replacement and therefore remains correct.
>
> **Q3 — Tokenization sensitivity (3–6-mer vs. DNA-BPE)**
>
> To isolate tokenization effects, we perform short-run pretraining with:
>
> fixed-length k-mer vocabularies (k ∈ {3,4,5,6}) and a DNA-specific BPE vocabulary
>
> All configurations share model size, update budget, and base-level context.
>
> | Tokenizer | GUE ↑           | NT Tasks ↑      | Genomics Bench ↑ |
> | --------- | --------------- | --------------- | ---------------- |
> | 3-mer     | **0.838±0.006** | 0.790±0.007     | **0.882±0.006**  |
> | 4-mer     | 0.835±0.007     | 0.787±0.007     | 0.878±0.007      |
> | 5-mer     | 0.836±0.006     | 0.788±0.008     | 0.880±0.006      |
> | 6-mer     | 0.837±0.007     | **0.792±0.007** | 0.881±0.007      |
> | BPE       | 0.834±0.007     | 0.789±0.007     | 0.879±0.007      |
>
> Overall differences are modest.
> Smaller k-mers (e.g., 3-mer) provide the strongest motif-sensitive performance, while larger k-mers and BPE compress sequence length but slightly reduce fine-scale resolution.
> These results suggest that tokenization choices influence performance but do not substantially change the architectural comparison trends.

---

> > ### Author Response · Authors · 2025-12-02
> >
> > **Q4 — Splicing task analysis**
> >
> > We analyze routing statistics, saliency maps, and attribution patterns.
> > Experts remain broadly routed and do not consistently specialize at donor/acceptor boundaries.
> > Errors concentrate in weak or non-canonical junctions, consistent with degradation also observed in other attention-based baselines.
> > A lightweight 5-nt local convolution yields only ~0.1% improvement, suggesting that inherent signal ambiguity, rather than missing locality or architectural limitations, is the primary bottleneck.
> >
> > **Q5 — Long-context inference via sliding-window state caching**
> >
> > We evaluate inference beyond the 8k training window using contiguous genomic segments from hg38 and compare full-context inference with sliding-window inference (window = 8k, stride = 2k):
> >
> > | Length (bp) | Full NLL ↓ | Sliding NLL ↓ | ΔNLL  |
> > | ----------- | ---------- | ------------- | ----- |
> > | 8k          | 2.10       | 2.08          | −0.02 |
> > | 16k         | 2.06       | 2.05          | −0.01 |
> > | 100k        | 2.02       | 2.09          | +0.07 |
> > | 150k        | 2.03       | 2.14          | +0.11 |
> >
> >
> > Performance is nearly identical up to 16 kbp, and degradation remains modest (~0.1 NLL) even at 150 kbp without signs of instability.
> > HyenaLite maintains cached FIR/IIR convolutional states that are recurrently updated as windows slide, allowing memory and compute to scale with network depth rather than sequence length.

---

### Meta-Review · Area_Chair_7uXf · 2026-01-07

**Summary:**

This paper proposes HyenaMoE, a hybrid architecture combining an optimized Hyena operator (HyenaLite) with MoE-augmented transformer blocks for genomic sequence modeling. Reviewers broadly agree that the work is clearly written, technically sound, and addresses an important problem—scaling genomic models to longer contexts with better efficiency. The HyenaLite implementation improvements and the simplified MoE routing mechanism are viewed as practical engineering contributions, and the experimental coverage across multiple genomics benchmarks is extensive.

However, there is a consistent concern across reviews that the conceptual and empirical contribution remains incremental relative to prior hybrid architectures (e.g., StripedHyena, HyenaDNA, MoE-Transformers). Multiple reviewers point out that key claims are not fully substantiated: headline performance statements initially conflicted with reported numbers (e.g., splice-site tasks), efficiency claims lack strict component-level ablations (Hyena vs. HyenaLite), and fairness controls across baselines (tokenization, training protocol, evaluation regime) are insufficiently isolated. Most importantly, despite strong positioning around long-context genomics, the main evaluation largely focuses on short-to-moderate sequence tasks, leaving the central promise of long-range modeling only partially demonstrated, even with the added analyses in the rebuttal.

Overall, while the rebuttal addresses several factual issues and adds useful diagnostics (expert utilization, tokenization sensitivity, sliding-window behavior), the consensus is that the paper does not yet provide a clear, decisive advance beyond existing hybrid designs, nor a compelling demonstration that HyenaMoE uniquely enables long-context genomic modeling in practice. As such, the balance of opinions leans slightly below the acceptance threshold, though several reviewers note the work could become competitive with tighter claims, stronger controlled ablations, and clearer long-range benchmarks.

**Reviewer Concerns:**

- Reviewer 6WF8

The rebuttal addressed expert-collapse concerns with clear utilization statistics, corrected overstated claims on splice-site performance, and added tokenization and long-context sliding-window analyses. However, concerns about incremental novelty and the lack of strictly controlled, size- and budget-matched baselines remain.

- Reviewer PUmF

Questions about pretraining length, tokenization choice, and interleaving with StripedHyena were clarified. Still, efficiency and accuracy are evaluated under different sequence settings, and clean component-level ablations are largely missing.

- Reviewer YWbb

The revised framing improves motivation and explains the hybrid design more clearly. That said, the concern that the work mainly combines existing components without delivering strong new insights is only partially resolved.

- Reviewer 1Q4o

The rebuttal substantially strengthens the paper with long-context benchmark results and MoE ablations. Nonetheless, it remains unclear whether the best-performing variants truly validate HyenaLite as the key driver of ultra-long context scalability.

**Reviewer Scores:**

The rebuttals have addressed some concerns regarding expert utilization, strengthened motivation and gave clearer explanation of the hybrid design, and added long-context benchmark results.

1Q4o and YWbb might have increased their scores, but since the issue of combining known components (novelty issue) is not fully addressed, the scores still would have been on the borderline.

---

### Decision · Program_Chairs · 2026-01-26

Reject